# Water Vapour and Methane Coupling in the Stratosphere observed using SCIAMACHY Solar Occultation Measurements

Stefan Noël, Katja Weigel, Klaus Bramstedt, Alexei Rozanov, Mark Weber, Heinrich Bovensmann, and John P. Burrows

Institute of Environmental Physics, University of Bremen, FB 1, P.O. Box 330440, 28334 Bremen, Germany

*Correspondence to:* S. Noël (stefan.noel@iup.physik.uni-bremen.de)

**Abstract.**

An improved stratospheric water vapour data set has been retrieved from SCIAMACHY/ENVISAT solar occultation measurements. It is similar to that successfully applied to methane and carbon dioxide. There is now a consistent set of data products for the three constituents covering the altitudes 17–45 km, the latitude range between about 50 and 70°N, and the period August 2002 to April 2012.

The new water vapour concentration profiles agree with collocated results from ACE-FTS and MLS/Aura to within ~5%. A significant positive linear change in water vapour for the time 2003–2011 is observed at lower stratospheric altitudes with a value of about $0.015 \pm 0.008 \, \mathrm{ppmv \, year^{-1}}$ around 17 km. Between 30 and 37 km the changes become significantly negative (about $-0.01 \pm 0.008 \, \mathrm{ppmv \, year^{-1}}$); all errors are $2\sigma$ values.

The combined analysis of the SCIAMACHY methane and water vapour time series shows the expected anti-correlation between stratospheric methane and water vapour and a clear temporal variation related to the Quasi-Biennial-Oscillation (QBO). Above about 20 km most of the additional water vapour is attributed to the oxidation of methane. In addition short-term fluctuations and longer-term variations on a timescale of 5–6 years are observed. The SCIAMACHY data confirm, that at lower altitudes the amount of water vapour and methane are transported from the tropics to higher latitudes via the shallow branch of the Brewer-Dobson circulation.

## 1 Introduction

Water vapour and methane play an important role in the chemistry of the stratosphere. For example the oxidation of methane generates the $HO_x$ radicals which catalytically destroy ozone ($O_3$) and are involved in many important stratospheric reactions. Water vapour is a key constituent of polar stratospheric clouds (PSCs) which play a unique role in the chemistry of the polar vortex, see e.g. Seinfeld and Pandis (2006). However, both water vapour and methane can also be used as dynamical tracers.

Methane is produced in the troposphere by various natural and anthropogenic emission processes. The identification of methane sources and sinks from the measurements of remote sensing instrumentation on satellites is currently an important research area (see e.g. Buchwitz et al., 2017, and references therein). The tropospheric lifetime of methane is about 10 years (Prinn et al., 2005). Consequently it is transported into the stratosphere.

The stratospheric entry value of water vapour is set by processes in the tropical tropopause layer (TTL; see e.g. Randel et al., 2004, Randel and Jensen, 2013, and references therein). There, the cold temperatures of the tropical tropopause result in a 'cold trap' (see e.g. Brewer, 1949, Holton and Gettelman, 2001, Read et al., 2004). A minimum in water vapour, which is around 2 km above the tropopause in the tropics, is called the hygropause. The level of minimum water varies with season and latitude. In the mid-latitudes, the level of the hygropause is a function of horizontal transport from the tropical cold point, so it will be elevated relative to the extratropical tropopause.

The water vapour, which enters the stratosphere through the TTL, is then transported via the Brewer-Dobson circulation from the tropics to higher latitudes. There are in principle two pathways for this transport (see e.g. Butchart, 2014, and references therein): At lower altitudes, air masses are transported via the shallow (or lower) branch of the Brewer-Dobson circulation. At higher altitudes the water vapour is transported by the deep (or upper) branch of the Brewer-Dobson circulation. This is illustrated in Fig. 1.

The amount of methane entering the stratosphere in the tropics depends on the changing strength of the sources (e.g. possible tropospheric trends). Variations in water vapour are caused by tropical tropopause temperature changes and dynamical effects, like changes in monsoon circulations, mixing in from mid-latitudes, convection, and microphysical processes. Furthermore, variations in the Brewer-Dobson circulation (on seasonal and inter-annual time scales) and the Quasi-Biennial-Oscillation (QBO, see e.g. Butchart, 2014 and references therein) play a role. The Brewer-Dobson circulation in the upper branch is driven by middle latitude planetary waves entering the stratosphere and as a consequence leads to adiabatic cooling in the tropical UTLS (upper troposphere / lower stratosphere region) related to the increased upwelling which strongly determines the stratospheric entry of water vapour in the tropics (Randel et al., 2006; Dhomse et al., 2008). As a component of the Brewer-Dobson circulation, the tropical upwelling is resposible for the transport of air masses from the troposphere into the stratosphere (both water vapor and methane) and influences the freeze-drying, i.e. the process through which the tropopause acts as a cold trap such that water vapour partly freezes out before reaching the stratosphere (e.g. Fueglistaler and Haynes, 2005).

Water vapour production in the stratosphere is largely a consequence of methane oxidation via the reaction

$$CH_4 + OH \rightarrow H_2O + CH_3 \tag{R1}$$

Rapid photochemical processes (see e.g. le Texier et al., 1988) result in the $CH_3$ being converted first to HCHO and then to $H_2O$ resulting in the net reaction:

$$CH_4 + 2O_2 \rightarrow 2H_2O + CO_2 \tag{R2}$$

For this overall reaction one methane molecule finally produces two water vapour molecules, which means that the sum of volume mixing ratios $[H_2O] + 2[CH_4]$, referred to as potential water (PW), see e.g. Rinsland et al. (1996); Nassar et al. (2005) and references therein, is expected to be roughly conserved following a stratospheric parcel. Since the actually conserved quantity is total hydrogen, this assumes that variations in $H_2$ can be neglected. The latter is in fact not always the case, as investigations by e.g. Juckes (2007) and Wrotny et al. (2010) indicate. Furthermore, potential water at a certain altitude may be affected by variations in the water vapour entry or QBO impacting lower stratospheric temperatures and mixing.

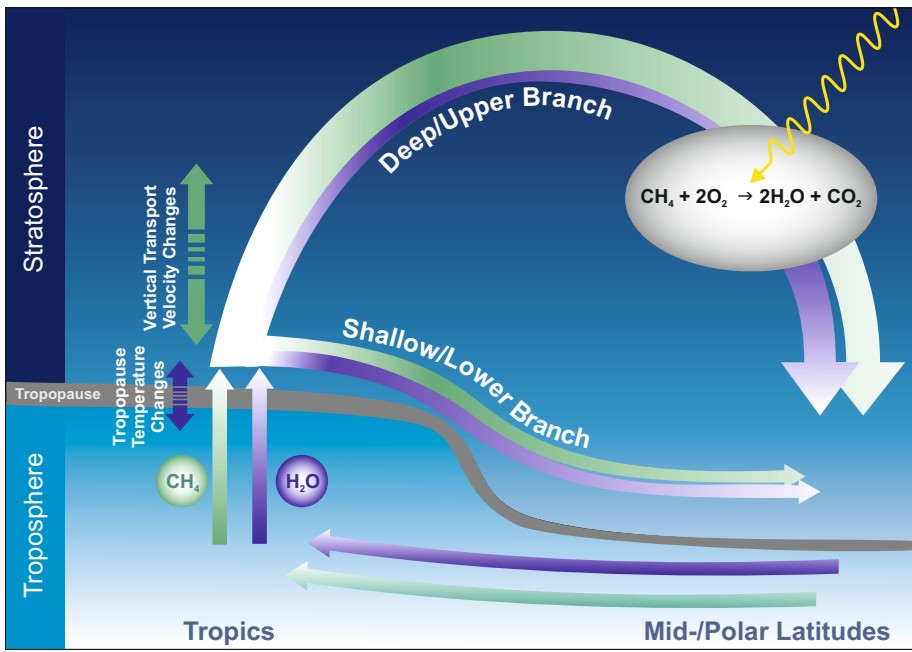

**Figure 1.** Simplified schematic view of transport pathways within the Brewer-Dobson circulation.

The rate determining step for the reaction (R2) is the photolysis of the different speciation in the reaction mechanism and thus depends on the availability of UV radiation. The reaction is therefore more effective higher in the stratosphere.

Another aspect to be considered is the transport time. The longer an air mass resides in the stratosphere, the more methane can be oxidised to water vapour. Measurements of age of air (see e.g. Haenel et al., 2015) indicate an increase of age of air with altitude at higher mid-latitudes from about 2 years at 15 km to about 8 years at 30 km. Age of air depends on season and latitude, and recent work by Ray et al. (2017) indicates, that these age estimates might be too high especially inside the polar vortex. Nevertheless, the methane–water vapour conversion process is expected to be more rapid and thus effective along the deep branch of the Brewer-Dobson circulation. However, the mixing of air masses during transport does not affect the total hydrogen balance such that potential water should still be conserved.

Simultaneous measurements of water vapour and methane data can therefore give information about sources and sinks of water vapour and dynamical effects in the stratosphere. This requires long-term data sets, which can be provided by satellite measurements.

For this, both water vapour and methane data should be collocated and accurate. If the underlying measurements are from the same instrument, the collocation of the two data sets is usually very close. Furthermore, possible systematic errors in methane or water vapour caused e.g. by instrument calibration or by the retrieval method may to some extent cancel for potential water.

Up to the present, data sets which fulfil these criteria are available only from a few instruments. This includes the Halogen Occultation Experiment (HALOE; Russell et al., 1993) on the Upper Atmospheric Research Satellite (UARS) measuring in solar occultation geometry from 1991 until 2005, see Rosenlof (2002), and the Atmospheric Chemistry Experiment Fourier

Transform Spectrometer (ACE-FTS) on SCISAT (Bernath et al., 2005) operating also in solar occultation geometry and providing scientific data since 2004. Methane and water vapour are two of the numerous ACE-FTS data products see e.g. Nassar et al. (2005). Stratospheric methane and water vapour profiles were also measured by the Michelson Interferometer for Passive Atmospheric Sounding (MIPAS; Fischer et al., 2008) on ENVISAT from 2002 to 2012 in limb geometry, see e.g. Payan et al. (2009); Laeng et al. (2015); Plieninger et al. (2016). Some early results from a combination of stratospheric methane and water vapour from MIPAS are given in Payne et al. (2005). Although primarily dedicated to measurements of polar mesospheric clouds (PMCs), the Aeronomy of Ice in the Mesosphere (AIM) Solar Occultation for Ice Experiment (SOFIE; Gordley et al., 2009) instrument also provides profiles of water vapour and methane. As part of the validation of the SOFIE V1.3 methane product, Rong et al. (2016) presented results from a combination of SOFIE and MIPAS methane with water vapour profiles from the Aura Microwave Limb Sounder (MLS; Waters et al., 2006).

The SCanning Imaging Absorption spectroMeter for Atmospheric CHartographY (SCIAMACHY; Bovensmann et al., 1999; Gottwald and Bovensmann, 2011) on ENVISAT performed measurements in various viewing geometries over a large spectral range from the UV to the SWIR. Among these are solar occultation measurements, which cover – depending on season – the spatial region between about 50°N and 70°N. Noël et al. (2016) presented an updated data set for stratospheric methane derived from SCIAMACHY solar occultation using the onion-peeling DOAS (ONPD) method. Already some years ago, Noël et al. (2010) showed first retrieval results for stratospheric water vapour profiles from SCIAMACHY which were based on a similar algorithm. Recently, the improved method used by Noël et al. (2016) has also been applied to water vapour, resulting in a consistent set of SCIAMACHY stratospheric water vapour and methane data.

In this manuscript, we shortly describe the updated water vapour algorithm in section 2. We then present the new water vapour results in section 3, which also includes a first validation by comparison with independent data sets and a combination of the new water vapour data with the methane data from Noël et al. (2016). The results are discussed in section 4. The conclusions are then presented in section 5.

## 2   H₂O Retrieval

The retrieval method used in this study is essentially the same as described in Noël et al. (2016), therefore only the principle idea is explained here.

We use transmission spectra as function of viewing (tangent) altitude derived from SCIAMACHY solar occultation measurements. For the water vapour retrieval, we take data in the spectral range 928 nm to 968 nm. The ONPD retrieval is then based on a combination of a weighting function DOAS fit (see e.g. Perner and Platt, 1979; Burrows et al., 1999; Coldewey-Egbers et al., 2005) with a classical onion peeling method (see e.g. Russell and Drayson, 1972). The retrieval altitude grid is 0 to 50 km in 1 km steps. The measured spectra are interpolated to this grid. The analysis starts at the top level and then proceeds downwards, taking into account the results from the upper levels. At each level, we determine the water vapour density from the difference between the measured transmission and a modelled one. This is done by fitting to the data a set of factors describing the change of an atmospheric parameter in combination with corresponding weighting functions. Such a weighting function

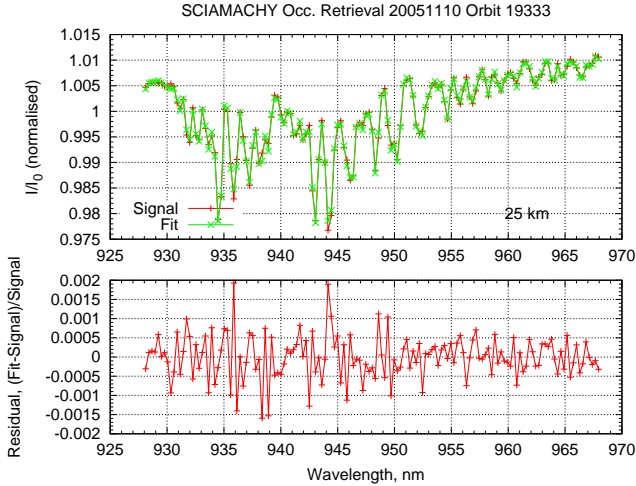

**Figure 2.** Example of a spectral fit. Top: normalised measured spectrum (red line) and fitted spectrum (green line) at 25 km tangent altitude. Bottom: resulting residual, i.e. relative difference between measurement and fit.

describes the change of the spectrum for a given change in a selected parameter, e.g. the water vapour concentration at this altitude. In the present case we consider in addition to water vapour also changes in ozone (which also absorbs in the spectral window used). The pressure and temperature profiles used in the study have been taken from ECMWF ERA Interim data (Dee et al., 2011). The related weighting functions have been determined from radiative transfer calculations using the SCIATRAN

model (Rozanov et al., 2014).

To account for spectrally broadband effects resulting from e.g. aerosols we also fit a polynomial to the spectra. A possible misalignment of the wavelength axis of the measured data is accounted for by fitting shift and squeeze parameters.

An example for the results of the fitting procedure is shown in Fig. 2. As can be seen, the measured transmissions is reproduced within an error of about 0.1%.

After the retrieval several additional corrections are performed as described in Noël et al. (2016):

- The retrieved profiles are smoothed with a 4.3 km boxcar to account for the vertical resolution of the measurements and to reduce oscillations in the retrieved number densities.

- Additional correction factors are applied for non-linearity and saturation effects (due to the limited spectral resolution of the measurements).

- The resulting errors are multiplied by a factor of 0.66 to correct for correlations between different layers not considered in the fit (see Noël et al., 2016, for details).

The resulting number density profiles are converted to volume mixing ratios (VMRs) using ECMWF pressure and temperature. The useful vertical range of the SCIAMACHY ONPD data is currently considered to be 17 to 45 km, mainly limited

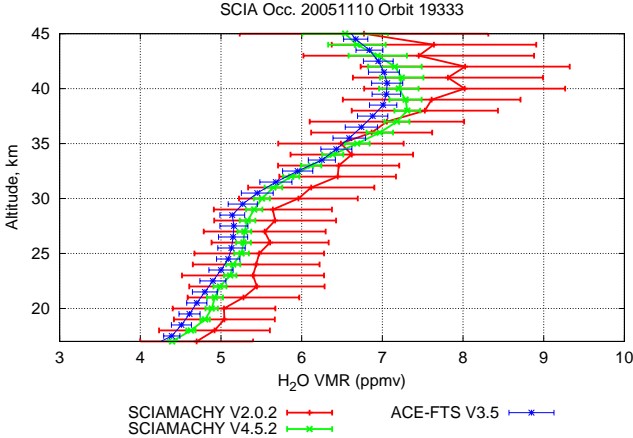

**Figure 3.** Example for $H_2O$ VMR profiles. Red: previous product (V2.0.2) from Noël et al. (2010). Green: current product (V4.5.2). Blue: collocated profile from ACE-FTS V3.5.

by noise and numerical effects at the upper altitudes and by tropospheric effects (e.g. clouds and increased refraction)) at the lower altitudes.

## 3   Results

### 3.1   $H_2O$ example data

Fig. 3 shows as an example the resulting water vapour VMR profile from a SCIAMACHY occultation measurement in November 2005. In green the result of the updated retrieval (V4.5.2) is shown. For comparison, the corresponding profile derived with the Noël et al. (2010) algorithm (V2.0.2) is plotted in red, and a collocated ACE-FTS profile (V3.5) in blue. The error bars denote the errors given in the products. Obviously, the new SCIAMACHY product is closer to the ACE-FTS results and the reported error is reduced compared to the older version. This is due to the improved retrieval method as described in Noël et al.

(2016). The most relevant changes are:

- Use of a weighting function DOAS based fit at each altitude.

- Better consideration of altitudes below the actual tangent height.

- Improved selection of measurements.

- Use of improved input spectral data (better pointing information and calibration).

- Use of an updated radiative transfer model (SCIATRAN V3).

- Updated error calculation.

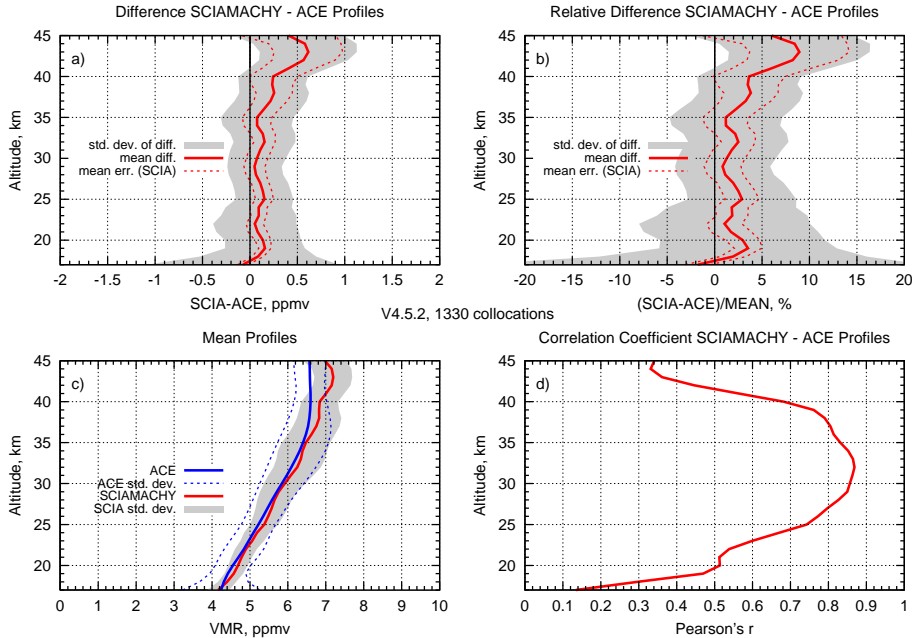

**Figure 4.** Comparison of retrieved SCIAMACHY $H_2O$ profiles with ACE-FTS data 2004–2012. **(a)** Mean difference plus/minus one standard deviation (shaded area) and mean error of SCIAMACHY data (dotted line). **(b)** Mean relative difference plus/minus one standard deviation (shaded area) and mean relative error of SCIAMACHY data (dotted line). **(c)** Mean profiles and standard deviations (red: SCIAMACHY, blue: ACE-FTS). **(d)** Correlation between SCIAMACHY and ACE-FTS data.

## 3.2 $H_2O$ validation

A large number of water vapour data products have been used in the analyses contributing to the second SPARC (Stratosphere-troposphere Processes And their Role in Climate) water vapour assessment (WAVAS-II; see e.g. Lossow et al., 2017, further publications in preparation). One activity of WAVAS-II was the inter-comparison of the different data sets, including a prelim-
5   inary earlier version (V4.2.1) of the SCIAMACHY ONPD product. The performance of the V4.2.1 product is very similar to the V4.5.2 product described in this manuscript. Consequently, in this section comparisons with collocated ACE-FTS (see e.g. Nassar et al., 2005) and MLS (see e.g. Carr et al., 1995; Lambert et al., 2007) data have been the focus of the additional valida-tion. In both cases the spatial collocation criterion is 800 km. For the ACE-FTS dataset we use only sunset data, as a result the local time difference to the SCIAMACHY data is usually less than one hour. For MLS we use a maximum time difference of 9
10  hours between the MLS and the SCIAMACHY measurements and always take the spatially closest match. Overall, this results in 1330 collocations with ACE-FTS data products and almost 35000 collocations with MLS data products between 2004 and 2012.

    Fig. 4 shows the results of the comparison between the SCIAMACHY ONPD V4.5.2 water vapour profiles and ACE-FTS V3.5 data. The MLS results are displayed in Fig. 5. The SCIAMACHY water vapour profiles agree with both data sets within

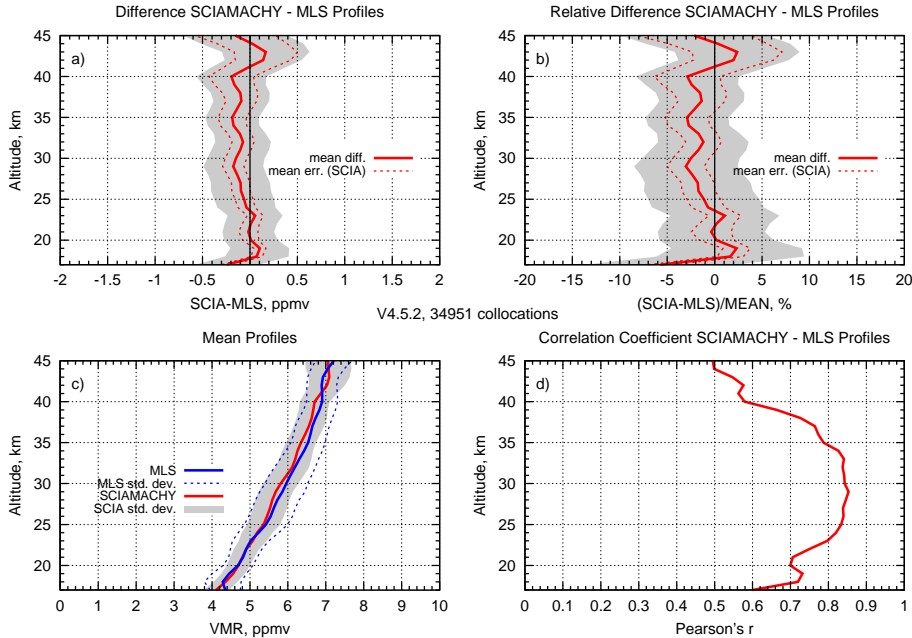

**Figure 5.** Same as Fig. 4, but for comparison of retrieved SCIAMACHY $H_2O$ profiles with MLS V4.2 data 2004–2012.

less than 5%. The SCIAMACHY water vapour VMRs are usually about 2–3% higher than those of ACE-FTS, but (except for the lowest altitudes) typically 2–3% smaller than MLS VMRs. A small vertical oscillation of 1–2% amplitude is observed in the differences. This is attributed to the SCIAMACHY data and is probably a retrieval artifact which was also seen in the SCIAMACHY ONPD methane and $CO_2$ data (Noël et al., 2016). The observed deviations are significantly smaller than the
typical error on the data products.

The correlation between SCIAMACHY and both ACE-FTS and MLS data is generally high (reaching about 0.85 at 30 km), but is poorer at lower and higher altitudes. The reduction at higher altitudes may be a consequence of the larger relative errors of the SCIAMACHY data, but as yet there is no clear explanation. At lower altitudes, differences in the variability of the data play a role, as can be inferred from the standard deviations shown in panels c) of Fig. 4 and 5. High correlation is achieved
when variability and variance are similar for both data sets, i.e. in this case both instruments see the same atmospheric changes.

### 3.3 Time series

The ONPD algorithm for water vapour has been applied to the entire set of SCIAMACHY measurements from August 2002 to April 2012. From the individual VMR profiles daily averages have been computed which are shown in Fig. 6 as function of time and altitude. As can be seen from the top curve in this figure, the latitude of the observation and the time in the year are
coupled. Observations in summer are typically at lower latitudes than in winter. This pattern is a result of the sun synchronous orbit of ENVISAT and the changing location of the solar occultation as a function of season. The average tropopause height,

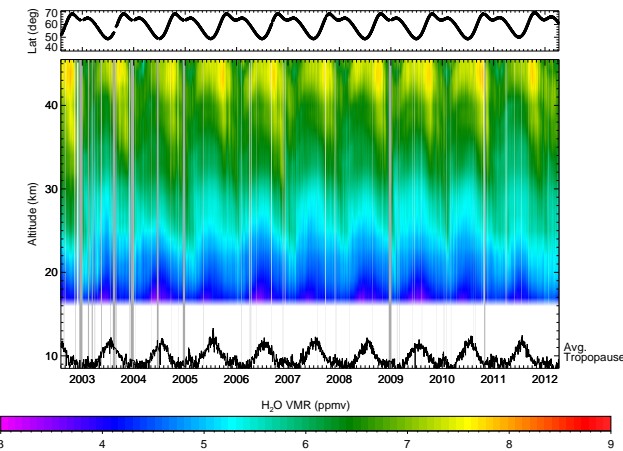

**Figure 6.** Time series of daily averaged SCIAMACHY $H_2O$ VMR profiles from August 2002 to April 2012. In the top graph the latitudes of observations as function of time are shown. Grey vertical bars mask out times of reduced SCIAMACHY performance or missing data. The black curve at lower altitudes shows the average tropopause height.

derived from collocated ECMWF data and shown by the black line near the bottom, varies in a similar way. The SCIAMACHY solar occultation data have therefore a specific temporal and spatial sampling.

The SCIAMACHY water vapour profiles behave in general as expected: Highest VMRs (up to about 8 ppmv) occur at high altitudes, lowest VMRs at lower altitudes. The variation with time follows roughly the tropopause / latitude pattern.

For a more detailed investigation of the observed behaviour of the water vapour and methane data products, we computed monthly anomalies from the SCIAMACHY $H_2O$ data in the same way as described Noël et al. (2016) and compared them with the $CH_4$ data from this study. This is achieved by first averaging the daily data over the months and then subtracting the long-term average for each month. To avoid different weighting of different months we limit this analysis to the time interval 2003 to 2011, i.e. we use only years for which data for all months are available.

In Fig. 7 the time series of the $H_2O$ and $CH_4$ anomalies are shown. There is a clear biennial structure visible in both of the data sets but of an opposite sign. As already mentioned in Noël et al. (2016), this structure is attributed to the Quasi-Biennial-Oscillation (QBO), see e.g. Baldwin et al. (2001).

The methane anomalies correspond to water vapour anomalies that are opposite in sign and twice the magnitude. This complies with the assumption, that most of the changes in water vapour are produced from methane via the net reaction (R2).

To investigate this further, Fig. 8 shows for some selected altitudes the water vapour anomalies as a function of time together with the methane anomalies multiplied by $-2$. If water vapour were produced solely via reaction (R2), both curves would be identical. This is to a good approximation the case for altitudes above about 25 km, where the water vapour variations follow quite well the methane variation. At 17 km, however, the methane anomaly does not vary much whereas the water vapour anomaly still shows a clear QBO signature, which is shifted in phase with respect to 25 km.

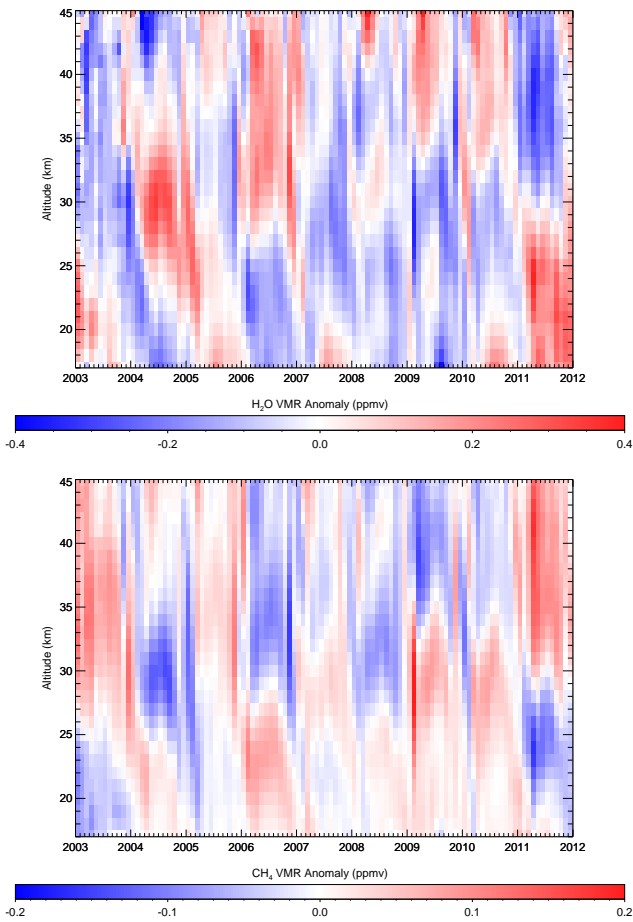

**Figure 7.** Time series of SCIAMACHY $H_2O$ (top) and $CH_4$ (bottom) monthly VMR anomaly profiles from January 2003 to December 2011. The $CH_4$ plot is taken from Noël et al. (2016). Note that for these data still the same latitudinal dependence as shown in Fig. 6 applies.

The dip in the water vapour anomalies at $17\,km$ in the middle of 2009 is related to the eruption of the Sarychev volcano on 12 June 2009, which reached these altitudes (Jégou et al., 2013). Note that this observed reduction of water vapour after the Sarychev eruption may be introduced by errors in the water vapour retrieval due to the remaining sensitivity of the retrieval method to aerosol. In the retrieval only spectrally broadband contributions of aerosols are considered, but there are also (second

5   order) effects e.g. caused by the vertical integration of the signal over the field of view, which may play a role in case of large aerosol concentrations. This issue is still under investigation.

The impact of the QBO is illustrated in Fig. 9 which shows SCIAMACHY methane and water vapour anomalies at $30\,km$ altitude as a function of time in comparison to the Singapore monthly mean stratospheric zonal wind at $10\,hPa$ (corresponding to about the same altitude), which is commonly used as index for the QBO (see e.g. Gebhardt et al., 2014). The Singapore wind

10   data have been provided by Freie Universität Berlin (2014). Negative wind direction corresponds to Easterly winds (marked

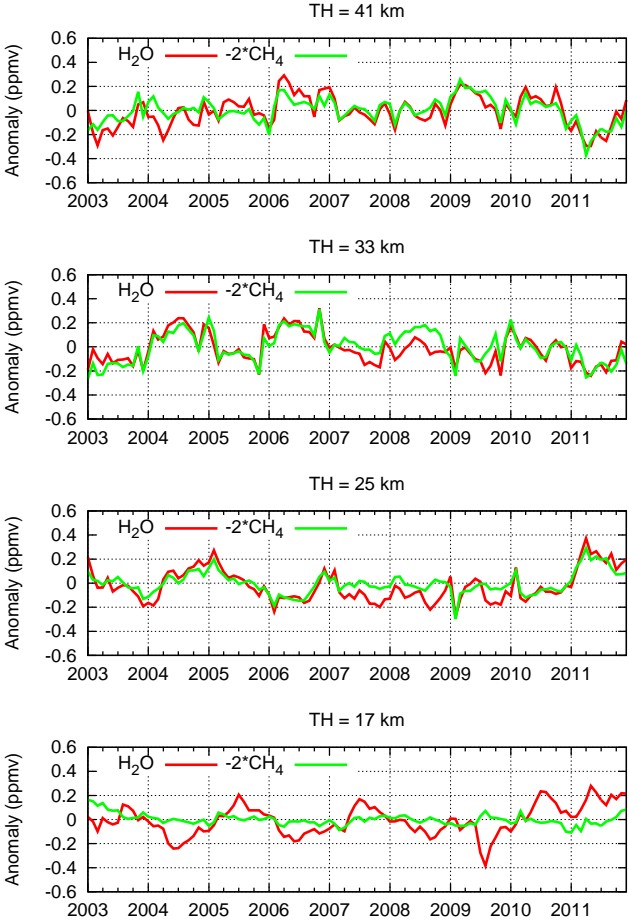

**Figure 8.** Time series of SCIAMACHY water vapour and methane anomalies at different altitudes. Methane data have been scaled by a factor $-2$.

blue in Fig. 9), positive direction to Westerly winds (marked red). Water vapour negative and positive anomalies are also plotted in blue and red, respectively. For the methane plot, the vertical axis and colouring has been inverted in order to take account of the production of water vapour from methane, where an increase in water vapour should correspond to a reduction of methane according to (R2).

5    Fig. 9 shows that water vapour and (inverted) methane anomalies follow the variation of the Singapore winds / QBO quite well, supporting the proposal that the changes are mainly affected by transport processes. The phase shift between stratospheric wind and SCIAMACHY data is caused by various dynamical processes during the transport of air from the tropics (where Singapore winds are measured) and the mid/high latitudes of the SCIAMACHY data and cannot be determined well from our 9-year time series. After about 2010 there are some differences between the wind data and the SCIAMACHY results. The

10    positive values in the wind data around 2010/2011 are hardly detected in the methane and water vapour data. On the other hand,

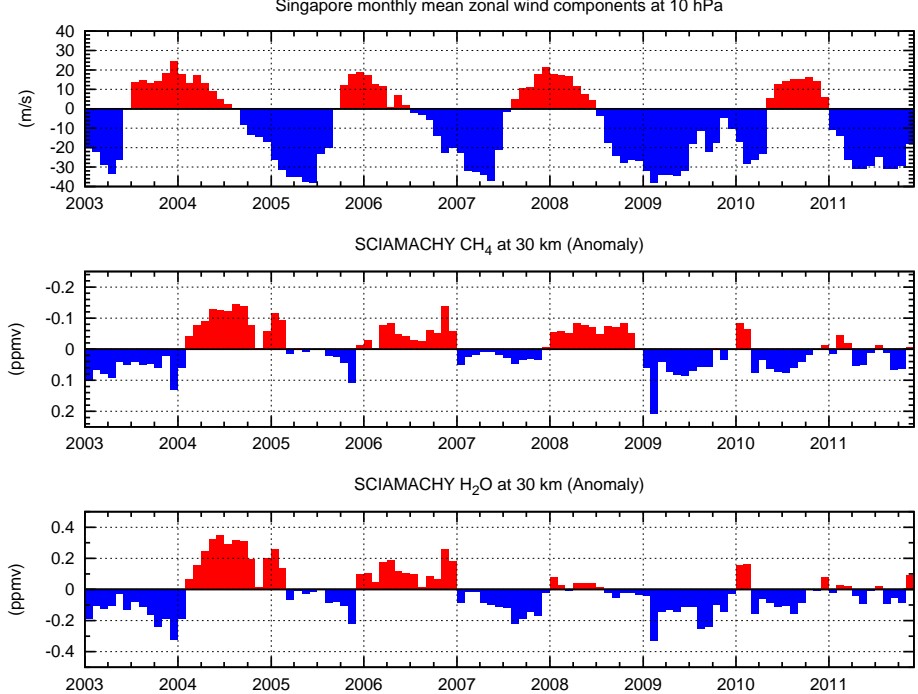

**Figure 9.** Time series of methane and water vapour anomalies at $30\,\mathrm{km}$ (middle and lower plots) and Singapore zonal wind at $10\,\mathrm{hPa}$, corresponding to about $30\,\mathrm{km}$ (top). Note that the vertical axis of the methane data is inverted and scaled differently than for water vapour.

positive anomalies of water vapour and (inverted) methane are quite strong at the begin of the time series. Possible reasons for these differences are currently unclear; maybe this is related to linear changes in the SCIAMACHY data (see below).

### 3.4 Potential water

To further investigate the production of water vapour from methane in the stratosphere a time series has been derived by adding the water vapour VMR anomalies to two times the methane VMR anomalies. As mentioned above this combination, referred to as potential water (Nassar et al., 2005), is assumed to be conserved if water vapour is solely produced from methane oxidation, and temporal variations of this quantity indicate changes in transport or additional sources and sinks. The result is displayed in Fig. 10.

Below about $20\,\mathrm{km}$ the biennial structure of the QBO is visible. After about 2010 there seems to be an additional increase of potential water, which is transported upwards. From the methane and water vapour time series shown in Fig. 8 it is evident that most of these changes are due to changing water vapour VMRs. The negative values in the second half of 2009 are associated with the Sarychev eruption, as explained before.

Between 20 and $40\,\mathrm{km}$ the vertical profile of the potential water anomaly is in summer (i.e. at lower latitudes) rather constant. In winter (corresponding to higher latitudes) sometimes larger variability is observed, possibly due to influences of the polar

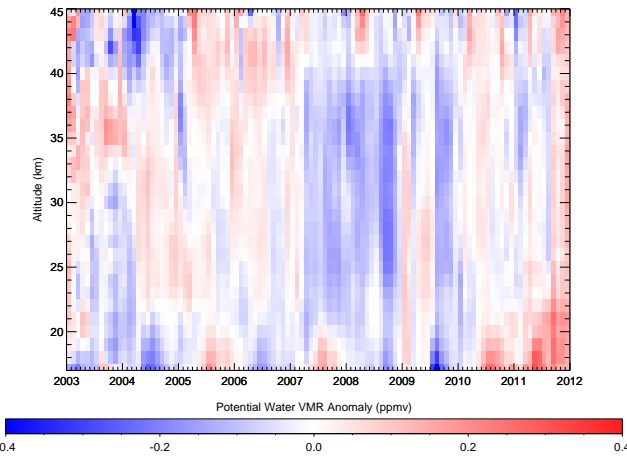

**Figure 10.** Potential water anomalies derived from combination of SCIAMACHY $H_2O$ and $CH_4$ anomalies (Fig. 7).

vortex. In 2003 and the first months of 2004, patterns are more patchy due to the different vertical sampling of the measurements at this time (see also Noël et al., 2016). In this time interval, positive anomalies occur around 35 km, negative anomalies above and below. Between about 2004 and 2007 potential water anomalies are typically positive whereas from 2007 to 2009 or 2010 they are mainly negative, and then later on in the time series they tend to be positive again. This implies a periodicity of about

5 to 6 years, but due to the limited length of the time series, this can only be confirmed in the future.

Above 40 km the variability of the potential water anomaly is quite high. This may be connected to the larger error and variance of the ONPD data at higher altitudes.

### 3.5   Linear Changes

The time series of SCIAMACHY data covers only ten (nine complete) years. Consequently it is not possible to derive from

these data long-term trends. Furthermore, the direct relation between the observational latitude and the time in the year of the SCIAMACHY measurements (see above) results in a particular spatial and temporal sampling. In this sense, the results shown in the following have to be interpreted as linear changes over the corresponding time interval 2003 to 2011 for a specifically sampled region between about 50 and 70°N.

To derive these changes, a linear regression has been fitted to the water vapour anomalies at each altitude similar to that used

in the earlier methane study, see Noël et al. (2016). For this, we take the anomaly times series at a selected altitude (see e.g. Fig. 8) and fit a straight line to it. The slope of this line is the estimated linear change for this altitude, the error of the linear change is the error of the slope given by the fit. This procedure is undertaken at each of the altitudes from 17 to 45 km, in 1 km steps. The resulting linear change profiles are displayed in Fig. 11.

The derived water vapour linear changes (left plot) are positive at altitudes below about 25 km, reaching a maximum value

of about $0.015 \pm 0.008$ ppmv year$^{-1}$ at 17 km. Between about 25 and 40 km the water vapour changes are negative and up

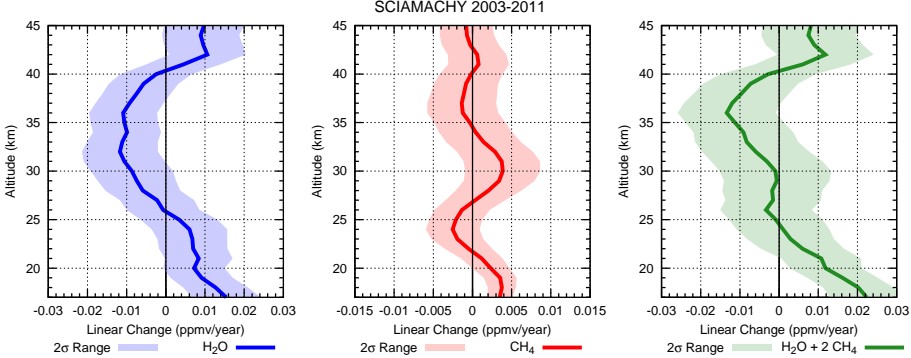

**Figure 11.** Calculated VMR linear changes of $H_2O$ (blue; left) and $CH_4$ (red; middle) from 2003 to 2011 as function of altitude. Methane data are from Noël et al. (2016). Right plot: Potential water linear changes derived from the combination of $H_2O$ and $CH_4$ linear changes.

to about -0.01±0.008 ppmv year$^{-1}$ (all errors are two standard deviations i.e. $2\sigma$ values). The $2\sigma$ error or uncertainty ranges also plotted indicate that the water vapour linear changes are not statistically significant at altitudes above 37 km and between 20 and 30 km where the change switches sign. A positive linear change in lower stratospheric water vapour during the time interval considered in this study has been observed by Urban et al. (2014) and Weigel et al. (2016) mainly in the tropics. As

already discussed in Noël et al. (2016) methane linear changes are also not significant except for the lowest altitudes, where they are in general agreement with tropospheric trends. However, it should be noted that errors of the data and autocorrelation of noise have not been considered in the linear changes fits, which might affect the resulting errors.

The potential water vapour linear change is the sum of the water vapour change and two times the methane change. This is an estimate for water vapour changes or methane changes not related to the stratospheric production of water vapour by methane.

If potential water is conserved, the potential water change should be zero. The potential water linear change profile is shown in the right plot of Fig. 11. The error of the potential water change has been derived via propagation of the errors of the methane and water vapour changes. Given that the changes in potential water between 21 and 45 km lack statistical significance, there is no evidence that water vapour is produced in the stratosphere by any mechanism other than methane oxidation. At the lower altitudes, a significant deviation of the potential water change from zero is observed (up to about 0.02±0.008 ppmv year$^{-1}$).

We have used a bootstrap method (see e.g. Efron, 1979) to investigate the sensitivity of our results to the chosen time interval. For this, we constructed for each altitude a set of 100 time series of same length as the original 2003–2011 time series (108 months) but consisting of randomly chosen combinations of results from individual months. For each of these artificial time series a linear change has been determined. For the resulting changes we determined for each altitude the mean linear change and the sample standard deviation. These values agree almost perfectly with the linear changes and their reported errors,

respectively, for the complete time series. This shows, that our results are robust within the given errors.

## 4 Discussion

The findings of this study are summarised as follows:

- Water vapour and methane time series and linear changes are different above and below about 20 km.

- At higher altitudes both water vapour and methane time series show a pronounced QBO signature.

- In the lower stratosphere, QBO signature is only visible in the water vapour data.

- There is a phase shift in the water vapour QBO signal between upper and lower altitudes.

- Potential water, the combination of methane and water vapour VMRs, is essentially conserved at higher stratospheric altitudes; the exceptions being some short-term events and a longer-term variation having patterns of about 5–6 years duration.

- The QBO signal is also visible in the potential water data at lower altitudes until about 2009/2010 after which potential water increases slowly.

These observations are consistent with a separation of the stratosphere into two vertical regimes. The lower regime is mainly affected by the shallow (or lower) branch of the Brewer-Dobson circulation Butchart (2014), whereas in the upper part the deep (or upper) branch of the Brewer-Dobson circulation dominates, see also Fig. 1. According to the data of the present study, this separation occurs at about 20 km. However it has to be kept in mind that this is an approximate value and that the vertical resolution of the SCIAMACHY solar occultation data is about 4 km.

In the lower regime, water vapour variability is mainly determined by variations due to the impact of the QBO and the Brewer-Dobson circulation on the tropopause temperature, see e.g. Fueglistaler and Haynes (2005). Methane entering the tropical stratosphere is mainly affected by tropospheric methane trends. In the lowermost extratropical stratosphere the water vapour and methane amounts follow the tropical amounts delayed by the transport time via the shallow branch of the Brewer-Dobson circulation.

The lack of a balance between the oxidation of methane and water vapour at lower altitudes is in fact not surprising, because the photochemical processes involved in the conversion of methane to water vapour are less effective there. This is because less UV radiation reaches these altitudes (le Texier et al., 1988). Furthermore, since the transport via the shallow branch is comparably fast (depending on latitude, altitude and season a few years or less from the entry point in the tropics to mid-latitudes, see Birner and Bönisch, 2011) the changes in water vapour and methane are not coupled in the extratropical lowermost stratosphere. This is the main reason why potential water is not conserved in this regime.

Schneising et al. (2011) estimated for the time interval 2007 to 2009 a tropospheric increase of methane of about $8 \, \mathrm{ppbv \, year^{-1}}$ following a period of no significant change from 2003 to 2007. Taking into account the delay between the tropospheric and a possible stratospheric trend related to the age of air (about 2–3 years between injection into the stratosphere at the tropics and measurement at 17 km at higher latitudes according to Haenel et al., 2015) explains part but not all of the increase of potential

water at lower altitudes after 2009/2010 shown in Fig. 10. It is therefore likely, that potential water at lower stratospheric altitudes is influenced by variations in the entry of tropospheric water vapour into the tropical stratosphere. Prior to the end of 2011 the positive potential water anomaly extends to higher altitudes. This is in agreement with the increasing age of air at higher altitudes.

In the upper dynamical regime water vapour is produced from methane oxidation and potential water anomalies are to a good approximation homogeneous with altitude and change on longer time scales. In the region of the deep branch of the Brewer-Dobson circulation, photochemical lifetimes decrease with altitude (le Texier et al., 1988). This enables oxidation of methane to water vapour to be completed rapidly. As a result variations of both gases are in phase and potential water is essentially conserved (Fig. 8). Consequently at these altitudes water vapour changes (relative to the injected amounts in the tropics) can
be concluded to be mainly determined by the oxidation of methane.

Another feature observed in the SCIAMACHY data between 2003 and 2011 is a change of potential water at higher altitudes on a timescale of 5–6 years. This could be attributed to low-frequency changes in the Brewer-Dobson circulation or long-term variations in water vapour trends currently under discussion (see e.g. Hegglin et al. (2014)).

A QBO signal is observed in both methane and water vapour at higher stratospheric altitudes. This QBO signature can be
explained by a QBO-dependent modulation of the transport to higher altitudes and to higher latitudes via the deep branch of the Brewer-Dobson circulation, similar to the variation in tropical aerosol extinction coefficients as seen by Brinkhoff et al. (2015) at 30 km. Randel et al. (1998) also observed a QBO signal in tropical methane from HALOE measurements on UARS above about 35 km but not below, correlated with the residual mean wind circulation. This is also in agreement with results from e.g. Niwano et al. (2003) and Minschwaner et al. (2016) who determined the vertical transport velocity in the tropics from HALOE
and MLS measurements, respectively, and confirmed a variation with QBO. The phase shift in the observed QBO signal of water vapour between 17 and 25 km (Fig. 8) is in agreement with the measurements of age of air by Haenel et al. (2015), which indicate differences in transport time.

## 5 Conclusions

A new stratospheric water vapour data set based on SCIAMACHY solar occultation measurements is made available. It covers
the latitude range between about 50 and 70°N and the altitude range from 17 to 45 km. It has been generated in a similar way to the methane product (Noël et al., 2016). Comparisons with independent data indicate the error of the water vapour profiles to be about 5%. Between 2003 and 2011 a significant positive linear change in water vapour is observed at altitudes below 20 km ($0.015\pm0.008$ ppmv year$^{-1}$ at 17 km). On the other hand, a significant negative water vapour linear change of about -0.01$\pm$0.008 ppmv year$^{-1}$ is derived for the altitude range 30–37 km; all errors are $2\sigma$ values.
The combination of the methane and water vapour time series data gives information about sources and transport of water vapour in the stratosphere.

At altitudes above about 20 km, variations in water vapour are clearly correlated with those of methane. A QBO signature is visible in both water vapour and methane anomaly time series, showing that transport from the tropics affects essentially the whole altitude range under investigation in this study.

The analysis of the combined water vapour and methane data sets reveals, that potential water, the sum of water vapour VMR and two time methane VMR, seems to be conserved between about 20 and 40–45 km. However, potential water is not constant over time. In addition to short term fluctuations a variation on a timescale of 5–6 years is observed.

At altitudes below about 20 km the QBO signature is only visible in water vapour but not in methane data. As a consequence, potential water also shows a significant QBO variation. In addition a continuous increase is observed after about 2009.

We explain this behaviour by a separation of the stratosphere into two regimes: i) altitudes above about 20 km being fed via the deep branch of the Brewer-Dobson circulation, and water vapour being produced from methane oxidation; ii) at altitudes below water vapour and methane have been transported from the tropics to higher latitudes via the shallow branch of the Brewer-Dobson circulation. The increase of tropospheric methane after 2007 reaches these lower stratospheric altitudes with a delay of about 2 years. This contributes in part to the observed increase of potential water after 2009, but additional processes such as changes of tropospheric water vapour input are required for a quantitative explanation.

*Data availability.* SCIAMACHY Level 1b data are available from ESA (https://earth.esa.int) after registration. All SCIAMACHY ONPD data V4.5.2 are available on request from S. Noël. The methane product V4.5.2 is also provided via the GHG-CCI web site http://www.esa-ghg-cci.org/ and accessible after registration.

*Competing interests.* The authors declare that they have no conflict of interest.

*Acknowledgements.* The Atmospheric Chemistry Experiment (ACE), also known as SCISAT, is a Canadian-led mission mainly supported by the Canadian Space Agency and the Natural Sciences and Engineering Research Council of Canada. The MLS data used in this research were produced by the Jet Propulsion Laboratory, California Institute of Technology under contract with the National Aeronautics and Space Adminstration (https://urs.earthdata.nasa.gov/, https://mls.jpl.nasa.gov/). The European Centre for Medium Range Weather Forecasts (ECMWF) provided the ERA Interim meteorological data used in this study. FU Berlin provided Singapore winds data. Geoffrey Toon of the NASA Jet Propulsion Laboratory is acknowledged for providing the empirical solar line list used in the retrieval. Part of this work was supported by the DFG Forschergruppe SHARP Water Vapor (SHARP-WV). This work has been funded by DLR Space Agency (Germany), the ESA GHG-CCI and by the University and the state of Bremen.

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
