# Peer review of "Water Vapour and Methane Coupling in the Stratosphere observed with SCIAMACHY Solar Occultation Measurements"

_Atmospheric Chemistry and Physics, 2017_

## Referee Comment (RC1) · Anonymous Referee #1 · 3 Nov 2017

This paper examines SCIAMACHY measurements of stratospheric water vapor and methane in the 50°N-70°N zonal band during 2002-2012. The focus is how stratospheric methane and water vapor abundances above about 20 km altitude are linked by the reactions that oxidize methane to water vapor. In this atmospheric region, the quantitative conversion of methane to water vapor makes total hydrogen ($H_2O+2*CH_4$) a conserved quantity as air masses mix and photochemically age. Inter-annual changes in $H_2O$ and $CH_4$ anomalies at various altitudes are attributed to the quasi-biennial oscillation (QBO). The data are also used to determine 9-year linear trends in $H_2O$, $CH_4$ and total hydrogen at stratospheric altitudes between 17 and 45 km. Most of the trends are not statistically significant. QBO effects on the relationship between $H_2O$ and $CH_4$

are discussed in the context of the upper and lower branches of the Brewer-Dobson circulation, for which the effects are different.

=============== General Comments ===============

The paper is generally well written, but some of the grammar and phrasing could be significantly improved by allowing a native English speaker to edit it.

The oxidation of hydrogen (H2) in the stratosphere is also a source of water vapor. How is it that the equation for total hydrogen ("potential water") does not include a term for H2 oxidation?

The paper would benefit from an early paragraph dedicated to describing the influences of the QBO on stratospheric entry mixing ratios of H2O and on the conversion of CH4 to H2O during transport from the tropics to higher latitudes. Currently there is a lot of attribution of inter-annual variations in H2O and CH4 to the QBO, but the actual QBO mechanisms that drive these variations are not really mentioned until late in the paper.

Similarly, an earlier introduction of concepts like the lower and upper branches of the Brewer-Dobson circulation, along with a description of mean age and its utility in discerning stratospheric transport pathways, would be very helpful when discussing the observed couplings between CH4 and H2O. Currently, these concepts are discussed too late in the paper.

The reasons why total hydrogen is a conserved quantity above the lowermost stratosphere may escape some readers. A simple explanation should be given, perhaps illustrating how mixing between air masses during transport does not change total hydrogen.

I don't see the need for Figure 11 as I am not quite sure what it explains. There is no caption to describe what is meant by the different shadings of green and purple (and white) arrows. I don't see anything in this Figure that isn't already described in detail in the text.

All trend values in the text should be presented with their uncertainties (95% confidence intervals) so the reader can gauge their significance. At many altitudes (if not all), the 9-year trends of CH4, H2O and total hydrogen are not statistically significant. Labeling trends as "negative", "positive", or "near-zero" is not justified if they are not statistically different from zero.

=============== Specific Comments ===============

Page 1, Line 4: "theses" should be "these"

P1, L6-8: Are these trends "significant" as stated? Please include their uncertainties to show that they are statistically different from zero.

P1, L10: "are strongly correlated" should be "are strongly anti-correlated"

Pages 1 and 8 have the QBO erroneously defined as the "Quasi-Biannual Oscillation" instead of "Quasi-Biennial Oscillation". You also use the term "bi-annual structure" on pages 8 and 11, where I assume you mean "biennial", since "bi-annual" means every 6 months.

P1, L11: Why does it only "seem" that most of the water vapor is produced by methane? What else might produce water vapor above 20 km?

P1, L13-15: Why should there be a "balance between water vapor and methane" at lower altitudes, unless of course the air masses came from higher in the stratosphere where there is a photochemical balance between CH4 and H2O? But here you mention only "the lower branch of the Brewer-Dobson circulation" where this photochemical balance does not exist. I don't understand the intention of this sentence and, to me, it is confusing.

P1, L17-18: It is too strong to say that these three greenhouse gases "determine the climate on our planet" when there are many, many contributors to Earth's climate.

P1, L22: If "methane is mainly produced in the troposphere", where else is it produced?

[Figure]

P1, L23: For decades, there have been attempts to produce spatially-resolved esti-mates of CH4 sources using globally-distributed surface measurements and inverse models. This is not something novel. How are satellite measurements used to identify methane sinks?

P2, L1: What is the "long" lifetime of "tropospheric methane"? Be more quantitative.

P2, L5: There were papers published long before 2001 that describe the "cold trap". In fact, there was some pioneering work performed back in the 1940s by Brewer and Dobson.

P2, L8: What is the connection between "tropical upwelling" and the "freeze-drying process"?

P2, L10: Why only "in the middle stratosphere and above" is water vapor "produced from (the) oxidation of stratospheric methane"? Both le Texier et al. (1988) and Rohs et al. (JGR, 2006) clearly show that some methane is oxidized in the lower stratosphere.

P2, L17: The concept of potential water, historically referred to as "total hydrogen", being conserved in stratospheric air masses as they mix and photochemically age, has been known for a long time. It pre-dates Rinsland et al. (GRL, 1996), so citing a 2005 paper here ignores the pioneering work on this topic that was performed well before the 21st century.

P2, L19-21: It is not "the combination", but rather "simultaneous measurements of" H2O and CH4 profiles, that are useful in understanding the connection between the two gases. Why is it best that they be measured by the same instrument? Does this improve the accuracy of H2O and CH4 retrievals, and therefore total hydrogen values?

P5, L9-11: A "criterium" is a bicycle race. Instead use "criterion" (singular form of criteria). I'm not sure what "a maximum time distance of 9 hours" means. And does "the closest match" refer to time or distance?

P5, L15-17: What version of MLS retrievals are you using? Hopefully the latest and

greatest, v4.2. The phrases "slightly higher" and "typically smaller" convey very little information. Please be more quantitative.

P6, Figure 3: I would be careful when using the term "absolute differences" because "absolute" may infer absolute values. Given the x-axis units (ppmv or %) I think it is safe to remove "absolute" and "relative" from the Figure headings.

P6, Figure 3d: Why does this vertical profile of correlation coefficients for SCIA vs ACE have such an altitude-dependent shape? The scatter in SCIA-ACE differences (ppmv and %) does increase somewhat near the lower and upper altitude boundaries, but is this enough to decrease the correlation coefficients near 17 and 45 km by more than a factor of two from those in the 25-40 km range? Do the correlation coefficients decline because of diminishing data populations as the altitude boundaries are approached? Figure 4d has a similar shape, but the r values don't fall so severely as the boundaries are approached. What makes these panel (d) curves similar in shape but so different in r values near the altitude boundaries?

P8, L3-5: What is meant by "bi-annual structure" in Figure 6? I don't see any cycles in the H2O or CH4 anomalies that clearly repeat with a 6-month (biannual) or 2-year (biennial) period. I do see lots of inter-annual variability. Is that what you want to say? Also, why does one expect inter-annual variability in CH4 and H2O because of the QBO? What are the mechanisms that drive changes in both?

P8, L6-8: "show an inverted behavior". Do you instead mean "opposite behavior" since "opposite" implies negative vs positive? The water vapor anomalies are "about twice as high negative" is awkwardly worded. How about "The methane anomalies correspond to water vapor anomalies that are opposite in sign and twice the magnitude." Also, the statement "that most of the water vapor is produced from methane" is not correct since mixing ratios of H2O are ~4 ppmv at stratospheric entry and ~7 ppmv at 45 km.

P9, L2-3: Don't the water vapor anomalies at 17 km also show year-to-year differences in the amount of water passing through the tropical cold trap, i.e., variability not related

to the QBO? By how many months is the QBO signal at 17 km "shifted in phase" from that at 25 km? Is the reason for this phase shift that the QBO propagates downward?

P9, L4: "downward peak" is contradictory. How about "dip"? I presume here you are still discussing the 17-km data?

P9, L10: I would call the Singapore zonal wind average a QBO "index" rather than a "proxy".

Figure 8 caption: Not only is the y-axis for CH4 inverted, it is also scaled differently than the y-axis for H2O.

P10, L6-8: What is the average transport time from the tropics to the northern 50-70° latitude at 30 km? You could determine this by independently regressing the H2O and CH4 anomalies against U10 and progressively delaying the U10 index one month at a time, finding the delay that produces the highest correlation coefficients. On Line 8 you say "positive anomaly in the wind data", but Figure 8 doesn't show wind anomalies. It is strange that the positive anomalies in H2O and –CH4 at the beginning of 2010 were not preceded by positive zonal mean winds.

P11, L1: Water vapor is also produced by the oxidation of hydrogen (H2) in the stratosphere. How does this factor into $H2O + 2*CH4$ = constant?

P11, L4-5: Why is the QBO signal visible only below 20 km in Figure 9? What mechanism alters $H2O + 2*CH4$ below 20 km but not above this altitude? Only Figure 7d shows greater variations in H2O anomalies than in CH4 anomalies.

P11, L11-13: What could possibly drive changes in $H2O + 2*CH4$ with a periodicity of 5-6 years? I don't think this statement is supported by Figure 9 that spans only 9 years.

P11, L15: I don't see the that scatter (std dev) in SCIAMACHY retrievals increases significantly above 40 km (Figures 3c and 4c), so what do you mean here by "large uncertainties of the ONPD data at higher altitudes"?

P11, L17-18: Please expand your description of the linear trend fitting here, at least in a general way. At what altitudes did you determine trends? Did you perform any vertical averaging (other than averaging kernels) of the profiles before determining trends? There is not enough information presented here to simply reference an earlier paper.

P11, L20: Here and throughout, all trends need to be presented with their 95% confidence intervals. Otherwise, the reader has no idea if the trends are statistically significant or not unless they check Figure 10. Many of the trends between 25 and 40 km are NOT negative, they are indistinguishable from zero. Only the $H_2O$ trends between 31 and 37 km can be labeled as negative.

Figure 10: Please scale the x-axis for $CH_4$ accordingly for $d[H_2O]/dt + 2*(d[CH_4]/dt) = 0$. Wouldn't one expect a positive trend in $CH_4$ accompanying the negative trend in $H_2O$ between 31 and 37 km?

P12, L7-8: Please remove one of the repeated "an estimate"

P12, L12: "not disproved" is a very weak way to say this. How about "Given that the trends in potential water between 21 and 45 km lack statistical significance, there is no evidence that water vapor is produced in the stratosphere by any mechanism other than methane oxidation."

P12, L13: "where the trend itself is close to zero" is not supported by the 95% confidence intervals of -0.015 to +0.014 ppmv/year in the 25-30 km altitude range.

P12, L19-20: Why is this? You haven't explained why the QBO might influence $H_2O$ but not $CH_4$ in the lower stratosphere. You also haven't explained why there should be a lag between QBO water vapor signals in the upper and lower stratosphere. Is it a difference in the mean ages of the air masses? It would be a good idea to introduce the concept of mean age early in this paper if you are going to discuss differences in the "phasing" of QBO-induced water vapor signals at different altitudes.

P13, L5-9: I think this explanation should appear earlier in the paper. This is not a

conclusion of the paper, it is information pertinent to the understanding of why QBO "signals" in H2O at different altitudes are present at different times.

P13, L11: "water vapor is mainly produced from methane oxidation". What else produces water vapor at these altitudes? Also, I think you need to have a definitive statement earlier in the paper that the oxidation of methane to water occurs predominantly in the tropical stratosphere and the fraction of methane converted to water increases with altitude.

P13, L15-17: Don't forget the main driver of variability in stratospheric H2O entry mixing ratios is the seasonal cycle of tropical tropopause temperatures. Also, ENSO can significantly influence water vapor input to the tropical lower stratosphere by affecting tropical tropopause temperatures and through convective activity. A lack of strong seasonal, QBO and ENSO influences on UTLS methane DOES explain the lack of CH4 variability at 17 km.

P14, L5-7: As per my previous comment about introducing the concept of mean age, here at the end of the paper is just such an introduction. I think the paper would benefit from this appearing much earlier.

P14, L10: This sentence makes it sound like CH4 was emitted at 17 km. And is mean age really the elapsed time from emission, including transport time from extra-tropical sources to the tropics?

P14, L17: The concept of "QBO signal has to be carried by methane" is an awkward way of explaining QBO influences on the oxidation of CH4 to H2O. If the QBO can alter the strength of the Brewer-Dobson circulation then it can also change the amount of CH4 oxidized to H2O during poleward transport. Transport times depend on the strength of the B-D circulation because this can also alter the path (i.e., stronger = higher path) and therefore the amount of CH4 oxidized to H2O. I think a paragraph early in the paper should be dedicated to HOW the QBO affects stratospheric transport and therefore the amount of CH4 converted to H2O during transport from the tropical

lower stratosphere to the higher latitudes of your data set.

P14, L29-30: Please include trend uncertainties with the trends.

P15, L1: "At altitudes above about 20 km, variations in water vapor ..."

P15, L6: Why is potential water not constant over time? Were there changes in the stratospheric entry mixing ratios of $H_2O$? Of $CH_4$? Of both?

---

## Referee Comment (RC2) · Anonymous Referee #3 · 20 Nov 2017

This study nicely presents the SCIAMACHY H2O and CH4 measurements and their relationship. The SCIAMACHY measurements are a very valuable addition to the available H2O and CH4 measurements in the middle atmosphere over the period 2003-2012, and the results shown here are scientifically valuable.

However, in much of the text the authors seem to be trying very hard to create a mystery where there is none. There is (1) a QBO signature in H2O crossing the tropical tropopause and (2) a QBO signal due to changes in transport (age-of-air) which causes a variation in the amount of CH4 that has been oxidized to produce H2O. The authors repeatedly overemphasize the importance of small tropospheric CH4 variations on the

observed variations in stratospheric H2O. While gradually increasing anthropogenic CH4 is a very important driver of long-term change in H2O, variations in CH4 entering the stratosphere are only marginally relevant to the variations observed in these measurements, which span a decade. Small changes in tropopause temperature are a far more important driver of interannual changes in H2O entering the stratosphere as has been shown by many authors (e.g. Dessler et al., JGR 2014).

Figure 11 is appropriate for a review paper on atmospheric dynamics, and might be appropriate if the authors were running a dynamical model to compare with their measurements, but it seems inappropriate here.

On page 14 line 7 they state: "This is not the case for methane, which could explain the missing QBO signature in the methane time series at 17km." There is no need for a "could" here. The H2O entering is governed by tropopause temperatures, and the CH4 is not.

In paragraph following this (and in the last sentence of the conclusion) they again try to overemphasize the importance of CH4. There is nothing inherently wrong with pointing out that changes in CH4 may play a small part in the observed changes of H2O, but an increase of 8 ppbv/yr in CH4 over 4 years would yield only at most ∼0.064 ppmv of H2O over 4 years. This looks small when compared to the observed variations in potential water, and if CH4 were the major driver of these variations potential water would not show decreases. Only finally, at the end of this paragraph, do the authors mention that: "However, from the current data set an additional influence of varying tropospheric water vapour input on the observed increase of potential water cannot be ruled out." This is certainly the primary driver of the variations in potential water, as is well understood. In the last sentence of the manuscript the authors again seem to only reluctantly accept that "possibly in combination with changes of water vapour" are important. Presumably this refers to changes in water vapour entering the stratosphere, but even that is not clear.

Then, in the final paragraph of the discussion they say: "A remaining open issue is the QBO signal observed in both methane and water vapour at higher stratospheric altitudes. . . . Therefore the QBO signal has to be carried by methane, but as can be seen at lower altitudes the methane entering the stratosphere is not varied by QBO." This is all well understood, as the authors finally admit in the second half of this paragraph.

The abstract is similarly unnecessarily confusing. First, the phrase "SCIAMACHY methane and water vapour time series reveals that stratospheric methane and water vapour are strongly correlated". The implication seems to be that this is a new result. Please rephrase this as "reveals [or, better yet, "shows"] the expected anticorrelation between methane and water vapour". The next sentence reads: "Above about 20km most of the water vapour seems to be produced by methane, but short-term fluctuations and a temporal variation on a scale of 5–6 years are observed." First, there is no reason for a "seems" here. The authors should be able to calculate how much of the observed water vapour is produced by methane. Secondly, I do not understand how the second part of this sentence follows from the first following a "but". I finally have to admit that I do not understand what new point the authors are trying to make in the last sentence of the abstract.

A few minor additional points in the text:

I don't understand the statement on page 2 line 19: "roughly conserved in the stratosphere if no changes in mixing of air masses occur". What does "changes in mixing of air masses" mean?

On page 9 line 6: "the remaining sensitivity of the retrieval method to aerosol" is rather a roundabout way of saying "errors in the water vapour retrieval due to aerosols". This is essentially what the authors say in the next line.

---

## Referee Comment (RC3) · Anonymous Referee #2 · 21 Nov 2017

This paper describes a water vapor data set derived from SCIAMACHY solar occultation measurements. It covers the altitude region from 17-45 km and the latitude region from 50-70N over the time period Aug 2002 to Apr 2012. The authors describe the method, the data set and then attempt trend analysis and describe the co-relationship between their CH4 and H2O data. I think a new data set is a valuable contribution, and the validation comparing to ACE and MLS is also valuable. The analysis of variations related to the QBO and discussion of the BDC is repeating work that has already been done, much going back to studies from measurements taken by UARS or LIMS/SAM. I think the paper could be significantly shortened into a data description/validation paper and much of the QBO and total hydrogen (or potential water) discussion eliminated.

[Figure]

General comment: please have the native English speaking co-author edit the text when revised.

Specific comments: Abstract, line 13-15, I would think that at lower altitude, water vapor is largely impacted by the stratospheric input value (so tropical tropopause temperatures). The "balance" hasn't had time to be established with young lower stratospheric air.

Page 1, Introduction, L17-18, the climate of the planet is determined by many factors, not just greenhouse gases. Please rewrite this sentence.

Page 2, line 3, the sentence "Most of the water vapour is of natural origin and located in the troposphere."and then change "It enters" to "Water vapor enters"

Page 2, line 8, I don't think this is an entirely accurate statement, in particular that the BDC controls the freeze drying process. The BDC is a zonally averaged construct, and freeze drying (and the associated microphysics) is a local process.

Page 5, figure 2; (and related text). Some descriptions as to what the improvements made in the algorithm between the 2010 product 2.0.2 and the current one is warranted (rather than simply referring to the 2016 methane paper).

Page 8, line 3&4..i think you mean biennial not bi-annual

Page 8, discussion of the "inverted behavior" (or anti-correlation) between water and methane. This is well known behavior and probably doesn't need the extensive following discussion regarding the QBO.

Page 11: line 14….you don't have a long enough time series to talk about 5-6 year oscillations, just delete that comment.

Page 12: trend discussion: the data set under consideration is just 10 years. That is not long enough to talk about trends. The so called trend noted on line 8 (Urban et al 2014) is really a step function like feature, not a trend. With 10 years, you can look at

interannual variability, and perhaps should stick to just that. Show a time series, not a linear trend.

Page 12, line 13 "an estimate" is duplicated

Page 12, line 14. It is not true to say that if potential water is conserved, the trend should be zero. You could have a trend in water vapor entry value, thereby allowing a potential water trend. You could also have a trend in the input of methane, again allowing a potential water trend.

Page 12: I really don't understand the point of this sentence "Considering this error, the combined trend above about 20 km is in a statistical sense not significant, meaning that the assumption that all water vapour is produced from methane via the net reaction (R2) is not disproved by the measurements." One should keep in mind that all water vapor is not produced from methane (ie, the average entry value is on the order of 3.5 ppmv, current methane is ∼1.8 ppmv, so if all were oxidized you could get a contributions of 3.6 ppmv, so at most you could get half of water vapor from methane. It may be that here the authors are trying to assess contribution to the trend. Rohs et al, 2006, JGR, determined for the 78-03 trend in stratospheric water vapor, only 25% can be due to a trend in methane. A similar analysis could be done here, for the SCIAMACHY period.

Page 14, line 26-30: this description of the processes going on is in error. In the upper altitudes, water vapor changes are anti correlated with methane, and simply reflect age of air variations; the QBO signal is not "carried by methane".

---

## Author Comment (AC1) · 19 Dec 2017

We thank the reviewer for the detailed comments which will help us to improve the paper. In the following, the original reviewer's comments are given in *italics*, our answer in normal font and the proposed updated text for the new version of the manuscript in **bold** font.

Note: It seems that line numbers in the comments refer to the manuscript before technical corrections, not the published discussion version.

[Figure]

**General Comments**

- *The paper is generally well written, but some of the grammar and phrasing could be significantly improved by allowing a native English speaker to edit it.*

  We will try to improve the English in the revised version and let our English co-author check it again. Therefore, the updated text might change slightly in the final revised version.

- *The oxidation of hydrogen (H2) in the stratosphere is also a source of water vapor. How is it that the equation for total hydrogen ("potential water") does not include a term for H2 oxidation?*

  We use the definition of Nassar et al. (2005) for potential water which does not include H2 (assuming that variations of H2 are small). This is mentioned in the introduction.

- *The paper would benefit from an early paragraph dedicated to describing the influences of the QBO on stratospheric entry mixing ratios of H2O and on the conversion of CH4 to H2O during transport from the tropics to higher latitudes. Currently there is a lot of attribution of inter-annual variations in H2O and CH4 to the QBO, but the actual QBO mechanisms that drive these variations are not really mentioned until late in the paper. Similarly, an earlier introduction of concepts like the lower and upper branches of the Brewer-Dobson circulation, along with a description of mean age and its utility in discerning stratospheric transport pathways, would be very helpful when discussing the observed couplings between CH4 and H2O. Currently, these concepts are discussed too late in the paper. The reasons why total hydrogen is a conserved quantity above the lowermost stratosphere may escape some readers. A simple explanation should be given, perhaps illustrating how mixing between air masses during transport does not change total hydrogen.*

Will will add a corresponding paragraph in the introduction, which will be largely re-written in the revised version.

- *I don't see the need for Figure 11 as I am not quite sure what it explains. There is no caption to describe what is meant by the different shadings of green and purple (and white) arrows. I don't see anything in this Figure that isn't already described in detail in the text.*

  Indeed, this figure does not contain additional information about the results, but we think it is helpful to visualise the different transport pathways and related processes. We therefore prefer to keep the figure in the manuscript, but move it to the introduction. Different shadings are mainly for artistic purpose and should illustrate dynamics.

- *All trend values in the text should be presented with their uncertainties (95% confidence intervals) so the reader can gauge their significance. At many altitudes (if not all), the 9-year trends of CH4, H2O and total hydrogen are not statistically significant. Labeling trends as "negative", "positive", or "near-zero" is not justified if they are not statistically different from zero.*

  We will add trend uncertainties ($2\sigma$ values) and also adapt the text accordingly (see answers to specific comments below).

**Specific Comments**

- *Page 1, Line 4: "theses" should be "these"*

  Will be corrected.

- *P1, L6-8: Are these trends "significant" as stated? Please include their uncertainties to show that they are statistically different from zero.*

The mentioned trends are significant. Uncertainties (about 0.008 ppm/year, see also Fig. 10) will be added.

- *P1, L10: "are strongly correlated" should be "are strongly anti-correlated"*

  Will be corrected.

- *Pages 1 and 8 have the QBO erroneously defined as the "Quasi-Biannual Oscillation" instead of "Quasi-Biennial Oscillation". You also use the term "bi-annual structure" on pages 8 and 11, where I assume you mean "biennial", since "bi-annual" means every 6 months.*

  The referee is absolutely right – it should be "biennial" in all cases, sorry for this mistake. We will correct this.

- *P1, L11: Why does it only "seem" that most of the water vapor is produced by methane? What else might produce water vapor above 20 km?*

  There are in fact other sources of stratospheric water vapour under discussion, e.g. from aviation or volcanoes. However, this is mainly relevant for the lowest parts of the stratosphere, therefore we will change this sentence to:

  **Above about 20 km most of the water vapour is attributed to the oxidation of methane.**

- *P1, L13-15: Why should there be a "balance between water vapor and methane" at lower altitudes, unless of course the air masses came from higher in the stratosphere where there is a photochemical balance between CH4 and H2O? But here you mention only "the lower branch of the Brewer-Dobson circulation" where this photochemical balance does not exist. I don't understand the intention of this sentence and, to me, it is confusing.*

  We agree that this may be confusing and will reformulate the sentence:
**The SCIAMACHY data confirm, that at lower altitudes the amount of water vapour and methane are transported from the tropics to higher latitudes via the shallow branch of the Brewer-Dobson circulation.**

- *P1, L17-18: It is too strong to say that these three greenhouse gases "determine the climate on our planet" when there are many, many contributors to Earth's climate.*

Agreed. We will reformulate this to:

**Water vapour ($H_2O$), methane ($CH_4$) and carbon dioxide ($CO_2$) are all greenhouse gases.**

- *P1, L22: If "methane is mainly produced in the troposphere", where else is it produced?*

There are indeed no known stratospheric sources of methane. We will therefore remove "mainly".

- *P1, L23: For decades, there have been attempts to produce spatially-resolved estimates of CH4 sources using globally-distributed surface measurements and inverse models. This is not something novel. How are satellite measurements used to identify methane sinks?*

Several satellite instruments (including SCIAMACHY, but also GOSAT and soon TROPOMI on Sentinel 5p) provide CH4 data, usually total columns determined from nadir measurements. These can be used in combination with inversion models to derive sources and sinks. The referee might have a look at the GHG-CCI web site (http://www.esa-ghg-cci.org) for more information about available data sets.

- *P2, L1: What is the "long" lifetime of "tropospheric methane"? Be more quantitative.*

The lifetime of tropospheric methane is about 10 years, we will mention that.

- *P2, L5: There were papers published long before 2001 that describe the "cold trap". In fact, there was some pioneering work performed back in the 1940s by Brewer and Dobson.*

  It is true that the "cold trap" has been discussed before 2001. We only wanted to give some example references here. We will add as additional (early) example the work by Brewer(1949), see reference in Holton and Gettelman (2001).

- *P2, L8: What is the connection between "tropical upwelling" and the "freeze-drying process"?*

  Tropical upwelling transports air masses from the troposphere into the stratosphere. As mentioned before, the tropopause acts as a cold trap such that water vapour partly freezes out before reaching the stratosphere, which is therefore dry compared to the troposphere. We will explain this in the revised version.

- *P2, L10: Why only "in the middle stratosphere and above" is water vapor "produced from (the) oxidation of stratospheric methane"? Both le Texier et al. (1988) and Rohs et al. (JGR, 2006) clearly show that some methane is oxidized in the lower stratosphere.*

  In the lower stratosphere oxidation of methane is not the only source of water vapour, there is e.g. also a tropospheric source (as we discuss in the present manuscript). However, as this sentence may be misleading, we will replace "in the middle stratosphere and above" by "in the stratosphere".

- *P2, L17: The concept of potential water, historically referred to as "total hydrogen", being conserved in stratospheric air masses as they mix and photochemically age, has been known for a long time. It pre-dates Rinsland et al. (GRL, 1996), so citing a 2005 paper here ignores the pioneering work on this topic that was performed well before the 21st century.*

The concept of potential water is indeed older than the mentioned publication from Nassar et al. (2005). However, Nassar et al. (2005) define the term "potential water" in contrast to "total hydrogen" (which includes H2) as we use it in our manuscript, therefore we cite this paper here, but we will also add the Rinsland reference.

- *P2, L19-21: It is not "the combination", but rather "simultaneous measurements of" H2O and CH4 profiles, that are useful in understanding the connection between the two gases. Why is it best that they be measured by the same instrument? Does this improve the accuracy of H2O and CH4 retrievals, and therefore total hydrogen values?*

If measurements from the same instrument (and similar retrievals) are used, possible systematic effects caused by the instrument or the retrieval method may cancel. This should improve the accuracy of the resulting potential water / total hydrogen.

We will reformulate this:

**Ideally, both water vapour and methane should be retrieved from measurements by the same instrument. In this case, the collocation of the two data sets is very close. Furthermore, possible systematic errors caused e.g. by instrument calibration or by the retrieval method may to some extent cancel.**

- *P5, L9-11: A "criterium" is a bicycle race. Instead use "criterion" (singular form of criteria). I'm not sure what "a maximum time distance of 9 hours" means. And does "the closest match" refer to time or distance?*

We will replace "criterium" by "criterion". "a maximum time distance of 9 hours" refers to the difference between the measurement times of the two instruments; "the closest match" refers to spatial distance. For clarification, we will reformulate the corresponding sentence:

**For MLS we use a maximum time distance of 9 hours between MLS and SCIAMACHY measurements and always take the spatially closest match.**

- *P5, L15-17: What version of MLS retrievals are you using? Hopefully the latest and greatest, v4.2. The phrases "slightly higher" and "typically smaller" convey very little information. Please be more quantitative.*

We indeed use MLS V4.2 and will mention this in the text and the related figure caption. We will also give quantitative numbers in the related sentence:

**The SCIAMACHY water vapour VMRs are usually about 2–3% higher than those of ACE-FTS, but (except for the lowest altitudes) typically 2–3% smaller than MLS VMRs.**

- *P6, Figure 3: I would be careful when using the term "absolute differences" because "absolute" may infer absolute values. Given the x-axis units (ppmv or %) I think it is safe to remove "absolute" and "relative" from the Figure headings.*

"absolute" might indeed be misleading as we show positive and negative values and can be removed. We will modify Figs. 3 & 4 and their captions accordingly. We see however no problem with "relative" and would prefer to keep this in order to better distinguish panels a) and b).

- *P6, Figure 3d: Why does this vertical profile of correlation coefficients for SCIA vs ACE have such an altitude-dependent shape? The scatter in SCIA-ACE differences (ppmv and %) does increase somewhat near the lower and upper altitude boundaries, but is this enough to decrease the correlation coefficients near 17 and 45 km by more than a factor of two from those in the 25-40 km range? Do the correlation coefficients decline because of diminishing data populations as the altitude boundaries are approached? Figure 4d has a similar shape, but the r values don't fall so severely as the boundaries are approached. What makes these panel (d) curves similar in shape but so different in r values near the altitude boundaries?*

The possible reason for the decreasing r in Fig. 3 at lower altitudes is that the variability of the ACE-FTS data is higher than for SCIAMACHY. This can be seen from the standard deviations shown in panel c). High correlation is achieved when variability (standard deviation) is similar for both data sets, i.e. in this case both instruments see the same atmospheric changes. MLS standard deviation is at lower altitudes closer to that of SCIAMACHY, therefore the correlation is higher.

We suggest to add the following text to explain this:

**The correlation between SCIAMACHY and both ACE-FTS and MLS data is generally high (reaching about 0.85 at 30km), but is poorer at lower and higher altitudes. The reduction at higher altitudes may be a consequence of the larger relative errors of the SCIAMACHY data, but as yet there is no clear explanation. At lower altitudes, differences in the variability of the data play a role, as can be inferred from the standard deviations shown in panels c) of Fig. 3 and 4. High correlation is achieved when variability and variance are similar for both data sets, i.e. in this case both instruments see the same atmospheric changes.**

- *P8, L3-5: What is meant by "bi-annual structure" in Figure 6? I don't see any cycles in the H2O or CH4 anomalies that clearly repeat with a 6-month (biannual) or 2-year (biennial) period. I do see lots of inter-annual variability. Is that what you want to say? Also, why does one expect inter-annual variability in CH4 and H2O because of the QBO? What are the mechanisms that drive changes in both?*

We mean "biennial structure", i.e. a variation with a 2-year period. This is seen especially at altitudes around 25–30 km where red and blue patterns repeat about every two years. This is seen more clearly in the following figures; related mechanisms are discussed in the "Discussions" section later.

- *P8, L6-8: "show an inverted behavior". Do you instead mean "opposite behavior" since "opposite" implies negative vs positive? The water vapor anomalies are*

*"about twice as high negative" is awkwardly worded. How about "The methane anomalies correspond to water vapor anomalies that are opposite in sign and twice the magnitude." Also, the statement "that most of the water vapor is produced from methane" is not correct since mixing ratios of H2O are ∼4 ppmv at stratospheric entry and ∼7 ppmv at 45 km.*

Yes, we mean "opposite" and will change the text. "most of the water vapor is produced from methane" actually refers to anomalies, i.e. changes in water vapour and methane, we will clarify that.

New text:

**The methane anomalies correspond to water vapour anomalies that are opposite in sign and twice the magnitude. This complies with the assumption, that most of the changes in water vapour are produced from methane via the net reaction (R2).**

- *P9, L2-3: Don't the water vapor anomalies at 17 km also show year-to-year differences in the amount of water passing through the tropical cold trap, i.e., variability not related to the QBO? By how many months is the QBO signal at 17 km "shifted in phase" from that at 25 km? Is the reason for this phase shift that the QBO propagates downward?*

Water vapour entering the stratosphere in the tropics varies also due to a combination of QBO and BDC effects. However, our measurements indicate that QBO effects dominate in this case. We try to explain this in the "Discussions" section later in the manuscript. According to our explanation, the air at 17 km is several years younger than the air above about 25 km (because of the different pathways of the Brewer-Dobson-Circulation). The phase shift between 17 and 25 km is therefore not only a few months but probably more than one 2-year period, and it is not possible to determine the exact value from our data. Above about 25 km there are indeed some indications for downward transport, as can be seen in the

slanted structures of the anomalies shown in Fig. 6.

We will add the following text for clarification:

**Note that the age of air at these altitudes may be up to about 8 years according to e.g. Haenel et al. (2015). Consequently, the actual phase shift is expected to be larger than one 2-year period of the QBO. It therefore cannot be determined well from our 9-year time series.**

- *P9, L4: "downward peak" is contradictory. How about "dip"? I presume here you are still discussing the 17-km data?*

We will reformulate the sentence to clarify this:

**The dip in the water vapour anomalies at 17 km in the middle of 2009 is related to the eruption of the Sarychev volcano...**

- *P9, L10: I would call the Singapore zonal wind average a QBO "index" rather than a "proxy".*

OK, will be changed.

- *Figure 8 caption: Not only is the y-axis for CH4 inverted, it is also scaled differently than the y-axis for H2O.*

The caption will be changed accordingly:

**Note that the vertical axis of the methane data is inverted and scaled differently than for water vapour.**

- *P10, L6-8: What is the average transport time from the tropics to the northern 50-70° latitude at 30 km? You could determine this by independently regressing the H2O and CH4 anomalies against U10 and progressively delaying the U10 index one month at a time, finding the delay that produces the highest correlation coefficients. On Line 8 you say "positive anomaly in the wind data", but Figure*

*8 doesn't show wind anomalies. It is strange that the positive anomalies in H2O and CH4 at the beginning of 2010 were not preceded by positive zonal mean winds.*

Age of air at these altitudes is about 8 years (see manuscript and above). Therefore the delay between tropospheric winds and stratospheric H2O or CH4 is more than one QBO period. We think the time series is too short and does not contain enough distinct features to determine these large delay times.

We will change "positive anomaly" to "positive values" as these are indeed no anomalies:

**The positive values in the wind data around 2010/2011 are hardly detected in the methane and water vapour data.**

The behaviour after 2010 is indeed strange and needs further investigations, as we mention in the text.

- *P11, L1: Water vapor is also produced by the oxidation of hydrogen (H2) in the stratosphere. How does this factor into H2O + 2\*CH4 = constant?*

Indeed, H2 needs to be considered in the sum as only total hydrogen is conserved. However, as mentioned in the introduction, for potential water we assume that H2 variations can be neglected.

- *P11, L4-5: Why is the QBO signal visible only below 20 km in Figure 9? What mechanism alters H2O + 2\*CH4 below 20 km but not above this altitude? Only Figure 7d shows greater variations in H2O anomalies than in CH4 anomalies.*

We discuss this in the "Discussions" section. The basic idea is that H2O at higher altitudes is produced from CH4 such that the combination does not show a QBO signal. At lower altitudes, H2O shows a QBO signal caused by variations due to QBO effects on tropopause temperature. CH4 transport into the stratosphere is

not affected by tropopause temperature changes and therefore does not show a QBO signal.

- *P11, L11-13: What could possibly drive changes in H2O + 2\*CH4 with a periodicity of 5-6 years? I don't think this statement is supported by Figure 9 that spans only 9 years.*

There is no explanation for this 5–6 years periodicity yet. Possible reasons are variations in the Brewer-Dobson circulation or changes in water vapour trends; we will mention this in the discussion. We also agree that it is difficult to tell if this periodicity is real from our data.

To clarify this we will reformulate this sentence to:

**This implies a periodicity of about 5 to 6 years, but due to the limited length of the time series, this can only be confirmed in the future.**

- *P11, L15: I don't see the that scatter (std dev) in SCIAMACHY retrievals increases significantly above 40 km (Figures 3c and 4c), so what do you mean here by "large uncertainties of the ONPD data at higher altitudes"?*

This refers to the (mean) error on the data which increases with altitude, see Figs. 3 & 4 panels a) and b).

- *P11, L17-18: Please expand your description of the linear trend fitting here, at least in a general way. At what altitudes did you determine trends? Did you perform any vertical averaging (other than averaging kernels) of the profiles before determining trends? There is not enough information presented here to simply reference an earlier paper.*

We will modify the text to describe the fitting procedure further:

**To derive these changes, a linear regression has been fitted to the water vapour anomalies at each altitude similar to that used in the earlier methane**

[Figure]

**study, see Noël et al. (2016). For this, we take the anomaly times series at a selected altitude (see e.g. Fig. 7) and fit a straight line to it. The slope of this line is the estimated trend for this altitude, the error of the trend is the error of the slope given by the fit. This procedure is undertaken at all altitudes from 17 to 45 km, in 1 km steps. The resulting trend profiles are displayed in Fig. 10.**

- *P11, L20: Here and throughout, all trends need to be presented with their 95% confidence intervals. Otherwise, the reader has no idea if the trends are statistically significant or not unless they check Figure 10. Many of the trends between 25 and 40 km are NOT negative, they are indistinguishable from zero. Only the H2O trends between 31 and 37 km can be labeled as negative.*

  The derived values of the H2O trends between about 25 and 40 km are negative, but it is true that some of these trends are not significant. We explain in the text which regions are significant and which are not. For further clarification, we will add the uncertainties to the trends mentioned explicitly in the text.

- *Figure 10: Please scale the x-axis for CH4 accordingly for d[H2O]/dt + 2\*(d[CH4]/dt) = 0. Wouldn't one expect a positive trend in CH4 accompanying the negative trend in H2O between 31 and 37 km?*

  We will scale the x-axis of the CH4 plot in Fig. 10 by a factor of 2. One would indeed expect a positive trend for CH4 between 31 and 37 km, but the resulting errors on the trends are high, so the CH4 trends and also the combined PW trends are not significant.

- *P12, L7-8: Please remove one of the repeated "an estimate"*

  OK.

- *P12, L12: "not disproved" is a very weak way to say this. How about "Given that the trends in potential water between 21 and 45 km lack statistical significance,*

*there is no evidence that water vapor is produced in the stratosphere by any mechanism other than methane oxidation."*

OK, will be changed.

- *P12, L13: "where the trend itself is close to zero" is not supported by the 95% confidence intervals of -0.015 to +0.014 ppmv/year in the 25-30 km altitude range.*

Agreed. Although the value is close to zero the trend is not significant, we will remove this sentence.

- *P12, L19-20: Why is this? You haven't explained why the QBO might influence H2O but not CH4 in the lower stratosphere. You also haven't explained why there should be a lag between QBO water vapor signals in the upper and lower stratosphere. Is it a difference in the mean ages of the air masses? It would be a good idea to introduce the concept of mean age early in this paper if you are going to discuss differences in the "phasing" of QBO-induced water vapor signals at different altitudes.*

An explanation for the observed features in this list is given in the subsequent paragraphs in the manuscript. We will reformulate this section and add some additional information about age of air in the introduction (see answer to general comments).

- *P13, L5-9: I think this explanation should appear earlier in the paper. This is not a conclusion of the paper, it is information pertinent to the understanding of why QBO "signals" in H2O at different altitudes are present at different times.*

We will add some information about the different branches in the introduction:

**There are in principle two pathways for this transport (see e.g. Butchart, 2014, and references therein): At lower altitudes, air masses are transported via the shallow (or lower) branch of the Brewer-Dobson circulation.**

**At higher altitudes the water vapour is transported by the deep (or upper) branch of the Brewer-Dobson circulation.**

- *P13, L11: "water vapor is mainly produced from methane oxidation". What else produces water vapor at these altitudes? Also, I think you need to have a definitive statement earlier in the paper that the oxidation of methane to water occurs predominantly in the tropical stratosphere and the fraction of methane converted to water increases with altitude.*

We will remove "mainly" and add some more information in the introduction.

- *P13, L15-17: Don't forget the main driver of variability in stratospheric H2O entry mixing ratios is the seasonal cycle of tropical tropopause temperatures. Also, ENSO can significantly influence water vapor input to the tropical lower stratosphere by affecting tropical tropopause temperatures and through convective activity. A lack of strong seasonal, QBO and ENSO influences on UTLS methane DOES explain the lack of CH4 variability at 17 km.*

Since we are looking at anomalies here, seasonal cycle effects should be removed. During the period of SCIAMACHY measurements there were no strong ENSO events, so this impact should be limited. Therefore we think that the missing QBO influence is a valid (and in this specific case sufficient) explanation for the lack of CH4 variability at 17 km.

- *P14, L5-7: As per my previous comment about introducing the concept of mean age, here at the end of the paper is just such an introduction. I think the paper would benefit from this appearing much earlier.*

We will add some sentences on age of air in the introduction (see answer to general comments).

- *P14, L10: This sentence makes it sound like CH4 was emitted at 17 km. And is mean age really the elapsed time from emission, including transport time from*

*extra-tropical sources to the tropics?*

The formulation is indeed misleading. We will change this to:

**about 2–3 years between injection into the stratosphere at the tropics and measurement at 17 km at higher latitudes**

- *P14, L17: The concept of "QBO signal has to be carried by methane" is an awkward way of explaining QBO influences on the oxidation of CH4 to H2O. If the QBO can alter the strength of the Brewer-Dobson circulation then it can also change the amount of CH4 oxidized to H2O during poleward transport. Transport times depend on the strength of the B-D circulation because this can also alter the path (i.e., stronger = higher path) and therefore the amount of CH4 oxidized to H2O. I think a paragraph early in the paper should be dedicated to HOW the QBO affects stratospheric transport and therefore the amount of CH4 converted to H2O during transport from the tropical lower stratosphere to the higher latitudes of your data set.*

We will add a corresponding part in the introduction, see answer to general comments.

- *P14, L29-30: Please include trend uncertainties with the trends.*

Will be done.

- *P15, L1: "At altitudes above about 20 km, variations in water vapor . . ."*

Will be added.

- *P15, L6: Why is potential water not constant over time? Were there changes in the stratospheric entry mixing ratios of H2O? Of CH4? Of both?*

Actually, we do not know the reasons why potential water varies on a timescale of 5–6 years, but we will mention possible reasons (low-frequency variations in the Brewer-Dobson circulation or in water vapour trends) in the discussion. Our

data set does not extend to the tropics, therefore we cannot infer changes of the entry mixing ratios.

---

## Author Comment (AC2) · 19 Dec 2017

We thank the reviewer for the comments and will consider them in the revised paper as described below. In the following, the original reviewer's comments are given in *italics*, our answer in normal font and the proposed updated text for the new version of the manuscript in **bold** font.

- *This study nicely presents the SCIAMACHY H2O and CH4 measurements and their relationship. The SCIAMACHY measurements are a very valuable addition to the available H2O and CH4 measurements in the middle atmosphere over the*

[Figure]

*period 2003-2012, and the results shown here are scientifically valuable. However, in much of the text the authors seem to be trying very hard to create a mystery where there is none. There is (1) a QBO signature in H2O crossing the tropical tropopause and (2) a QBO signal due to changes in transport (age-of-air) which causes a variation in the amount of CH4 that has been oxidized to produce H2O. The authors repeatedly overemphasize the importance of small tropospheric CH4 variations on the observed variations in stratospheric H2O. While gradually increasing anthropogenic CH4 is a very important driver of long-term change in H2O, variations in CH4 entering the stratosphere are only marginally relevant to the variations observed in these measurements, which span a decade. Small changes in tropopause temperature are a far more important driver of interannual changes in H2O entering the stratosphere as has been shown by many authors (e.g. Dessler et al., JGR 2014).*

We agree that some of the statements/formulations in the manuscript may be misleading. We do not aim to propose new dynamical processes or explanations. Our intention is to present the new SCIAMACHY H2O data set and show via the combination with CH4 that information about atmospheric dynamics can be derived. This is not necessarily new information, but it shows the usefulness of the SCIAMACHY data.

We will clarify this in the revised version of the manuscript (see answers to the following comments and also our answers to the comments of referee #1).

- *Figure 11 is appropriate for a review paper on atmospheric dynamics, and might be appropriate if the authors were running a dynamical model to compare with their measurements, but it seems inappropriate here.*

We agree that Figure 11 does not present any new results. However, it summarises the different dynamical processes discussed in the manuscript and is therefore considered as helpful especially for the non-expert reader. We therefore prefer to keep this figure in the manuscript but will move it to the (modified)

Printer-friendly version

[Figure]

introduction.

- *On page 14 line 7 they state: "This is not the case for methane, which could explain the missing QBO signature in the methane time series at 17km." There is no need for a "could" here. The H2O entering is governed by tropopause temperatures, and the CH4 is not.*

  Agreed. We will remove "could".

- *In paragraph following this (and in the last sentence of the conclusion) they again try to overemphasize the importance of CH4. There is nothing inherently wrong with pointing out that changes in CH4 may play a small part in the observed changes of H2O, but an increase of 8 ppbv/yr in CH4 over 4 years would yield only at most ∼0.064 ppmv of H2O over 4 years. This looks small when compared to the observed variations in potential water, and if CH4 were the major driver of these variations potential water would not show decreases. Only finally, at the end of this paragraph, do the authors mention that: "However, from the current data set an additional influence of varying tropospheric water vapour input on the observed increase of potential water cannot be ruled out." This is certainly the primary driver of the variations in potential water, as is well understood. In the last sentence of the manuscript the authors again seem to only reluctantly accept that "possibly in combination with changes of water vapour" are important. Presumably this refers to changes in water vapour entering the stratosphere, but even that is not clear.*

  Actually, the referee is right here. An increase in CH4 due to tropospheric trends alone cannot quantitatively explain the observed increase in potential water.

  We will therefore reformulate this part:

  **Schneising et al. (2011) estimated for the time interval 2007 to 2009 a tropospheric increase of methane of about 8 ppbv year$^{-1}$ following a period of no significant change from 2003 to 2007. Taking into account the delay**

**between the tropospheric and a possible stratospheric trend related to the age of air (about 2–3 years between injection into the stratosphere at the tropics and measurement at 17 km at higher latitudes according to Haenel et al., 2015) explains part but not all of the increase of potential water at lower altitudes after 2009/2010 shown in Fig. 9. An additional influence of varying tropical tropospheric water vapour on the observed increase of potential water is therefore likely. Prior to the end of 2011 the positive potential water anomaly extends to higher altitudes. This is in agreement with the increasing age of air at higher altitudes.**

The conclusions will also be adapted:

**The increase of tropospheric methane after 2007 reaches these lower stratospheric altitudes with a delay of about 2 years. This contributes in part to the observed increase of potential water after 2009, but additional processes such as changes of tropospheric water vapour input are required for a quantitative explanation.**

- *Then, in the final paragraph of the discussion they say: "A remaining open issue is the QBO signal observed in both methane and water vapour at higher stratospheric altitudes ... Therefore the QBO signal has to be carried by methane, but as can be seen at lower altitudes the methane entering the stratosphere is not varied by QBO." This is all well understood, as the authors finally admit in the second half of this paragraph.*

As requested by reviewer #1, the introduction of the revised paper will contain more information about known dynamical processes.

For clarification, we will also reformulate this part as follows:

**Above 20 km, in the region of the deep branch of the Brewer-Dobson circulation, air is older. This enables oxidation of methane to water vapour to be completed rapidly. As a result variations of both gases are in phase and**

[Figure]

**potential water is essentially conserved (Fig. 7). Consequently at these altitudes water vapour changes can be concluded to be determined by the oxidation of methane. The QBO signal is observed in both methane and water vapour at higher stratospheric altitudes. In contrast, the tropospheric methane entering the stratosphere via the lower branch of the Brewer-Dobson circulation is not impacted by the QBO at lower altitudes.**

- *The abstract is similarly unnecessarily confusing. First, the phrase "SCIAMACHY methane and water vapour time series reveals that stratospheric methane and water vapour are strongly correlated". The implication seems to be that this is a new result. Please rephrase this as "reveals [or, better yet, "shows"] the expected anticorrelation between methane and water vapour".*

We will rephrase this sentence as suggested:

**The combined analysis of the SCIAMACHY methane and water vapour time series shows the expected anti-correlation between stratospheric methane and water vapour and a clear temporal variation related to the Quasi-Biennial-Oscillation (QBO).**

- *The next sentence reads: "Above about 20km most of the water vapour seems to be produced by methane, but short-term fluctuations and a temporal variation on a scale of 5–6 years are observed." First, there is no reason for a "seems" here. The authors should be able to calculate how much of the observed water vapour is produced by methane. Secondly, I do not understand how the second part of this sentence follows from the first following a "but".*

This part of the abstract will be reformulated accordingly:

**Above about 20 km most of the water vapour is produced by methane. In addition, short-term fluctuations and a temporal variation on a scale of 5–6 years are observed.**

- *I finally have to admit that I do not understand what new point the authors are trying to make in the last sentence of the abstract.*

  There is indeed no new finding here. We only want to state here, that the described effects can be seen in the SCIAMACHY data.

  We will clarify this:

  **The SCIAMACHY data confirm, that at lower altitudes the amount of water vapour and methane are transported from the tropics to higher latitudes via the shallow branch of the Brewer-Dobson circulation. Further, the increasing methane input into the stratosphere due to the rise of tropospheric methane after 2007 may have contributed to the increased water vapour in the extratropical lower stratosphere as observed by SCIAMACHY.**

- *A few minor additional points in the text: I don't understand the statement on page 2 line 19: "roughly conserved in the stratosphere if no changes in mixing of air masses occur". What does "changes in mixing of air masses" mean?*

  This refers to additional production / loss processes others than via the net reaction (R2), like production of H2O by oxidation of other hydrocarbons, but these are indeed rather negligible (as e.g. stated by Nassar et al., 2005). We therefore will remove "if no changes in mixing of air masses occur".

- *On page 9 line 6: "the remaining sensitivity of the retrieval method to aerosol" is rather a roundabout way of saying "errors in the water vapour retrieval due to aerosols". This is essentially what the authors say in the next line.*

  To clarify this, we will reformulate this sentence as follows:

  **Note that this observed reduction of water vapour after the Sarychev eruption may be introduced by errors in the water vapour retrieval due to the remaining sensitivity of the retrieval method to aerosol.**

[Figure]

---

## Author Comment (AC3) · 19 Dec 2017

We thank the reviewer for the comments and will consider them in the revised paper as described below. In the following, the original reviewer's comments are given in *italics*, our answer in normal font and the proposed updated text for the new version of the manuscript in **bold** font.

[Figure]

**General comments**

- *This paper describes a water vapor data set derived from SCIAMACHY solar occultation measurements. It covers the altitude region from 17-45 km and the latitude region from 50-70N over the time period Aug 2002 to Apr 2012. The authors describe the method, the data set and then attempt trend analysis and describe the co-relationship between their CH4 and H2O data. I think a new data set is a valuable contribution, and the validation comparing to ACE and MLS is also valuable. The analysis of variations related to the QBO and discussion of the BDC is repeating work that has already been done, much going back to studies from measurements taken by UARS or LIMS/SAM. I think the paper could be significantly shortened into a data description/validation paper and much of the QBO and total hydrogen (or potential water) discussion eliminated.*

The aim of the paper is not only to present and validate the new SCIAMACHY H2O data set. We also want to show the usefulness of the H2O SCIAMACHY data in combination with other data, e.g. in the context of dynamical studies. The results obtained related to BDC or QBO are indeed not new, but we can confirm them with the new SCIAMACHY data. Therefore we prefer to keep the discussion on dynamical effects in the paper, but will clarify this in the revised version (see also answers to comments of other referees).

- *General comment: Please have the native English speaking co-author edit the text when revised.*

Will be done. Therefore, the updated text might change slightly in the final revised version.

**Specific comments:**

- *Abstract, line 13-15, I would think that at lower altitude, water vapor is largely impacted by the stratospheric input value (so tropical tropopause temperatures). The "balance" hasn't had time to be established with young lower stratospheric air.*

  Agreed, "balance" is misleading. We will reformulate this as follows:

  **The SCIAMACHY data confirm, that at lower altitudes the amount of water vapour and methane are transported from the tropics to higher latitudes via the shallow branch of the Brewer-Dobson circulation. Further, the increasing methane input into the stratosphere due to the rise of tropospheric methane after 2007 may contribute to the increased water vapour.**

- *Page 1, Introduction, L17-18, the climate of the planet is determined by many factors, not just greenhouse gases. Please rewrite this sentence.*

  Agreed. New text:

  **Water vapour (H2O), methane (CH4) and carbon dioxide (CO2) are all greenhouse gases.**

- *Page 2, line 3, the sentence "Most of the water vapour is of natural origin and located in the troposphere." and then change "It enters" to "Water vapor enters"*

  This part will be rewritten:

  **The amount of water vapour in the troposphere is very large compared with that in the rest of the atmosphere. Water vapour enters the stratosphere mainly through the tropical tropopause layer ...**

- *Page 2, line 8, I don't think this is an entirely accurate statement, in particular that the BDC controls the freeze drying process. The BDC is a zonally aver-*

*aged construct, and freeze drying (and the associated microphysics) is a local process.*

The term "controls" indeed might not be accurate here w.r.t. to freeze drying. We will reformulate the text as follows:

**The Brewer-Dobson circulation controls the tropical upwelling, i.e. the transport of air masses from the troposphere into the stratosphere (both water vapor and methane) and influences the freeze-drying, i.e. the process through which the tropopause acts as a cold trap such that water vapour partly freezes out before reaching the stratosphere.**

- *Page 5, figure 2; (and related text). Some descriptions as to what the improvements made in the algorithm between the 2010 product 2.0.2 and the current one is warranted (rather than simply referring to the 2016 methane paper).*

  We will add the following information:

  **This is due to the improved retrieval method as described in Noël et al. (2016). The most relevant changes are:**

  – **Use of a weighting function DOAS based fit at each altitude.**
  – **Better consideration of altitudes below the actual tangent height.**
  – **Improved selection of measurements.**
  – **Use of improved input spectral data (better pointing information and calibration).**
  – **Use of an updated radiative transfer model (SCIATRAN V3).**
  – **Updated error calculation.**

- *Page 8, line 3&4..i think you mean biennial not bi-annual*

  Indeed. Will be changed.

- *Page 8, discussion of the "inverted behavior" (or anti-correlation) between water and methane. This is well known behavior and probably doesn't need the extensive following discussion regarding the QBO.*

As mentioned above, we would like to keep this discussion on QBO in order to show the capabilities of the SCIAMACHY data.

- *Page 11: line 14. You don't have a long enough time series to talk about 5-6 year oscillations, just delete that comment.*

We agree that it is difficult to tell if this 5-6 year periodicity is real from our data, as we state in the text. To clarify this we will reformulate this sentence to:

**This implies a periodicity of about 5 to 6 years, but due to the limited length of the time series, this can only be confirmed in the future.**

- *Page 12: trend discussion: the data set under consideration is just 10 years. That is not long enough to talk about trends. The so called trend noted on line 8 (Urban et al 2014) is really a step function like feature, not a trend. With 10 years, you can look at interannual variability, and perhaps should stick to just that. Show a time series, not a linear trend.*

Indeed 10 years is too short for a trend in the climatological sense. Therefore, what we present here are essentially estimated changes over this time interval. Knowing their limitations, these changes can nevertheless provide interesting information. We therefore would like to keep the "trend" results in the paper, but we will add a clarification at the begin of the trends section:

**The time series of SCIAMACHY data covers only ten (nine complete) years. Consequently it is not possible to derive from these data long-term trends. In this sense, the trends shown in the following have to be interpreted as changes over the corresponding time interval 2003 to 2011. To derive these changes, a linear regression has been fitted to the water vapour anomalies ...**

- *Page 12, line 13 "an estimate" is duplicated*

  Will be removed.

- *Page 12, line 14. It is not true to say that if potential water is conserved, the trend should be zero. You could have a trend in water vapor entry value, thereby allowing a potential water trend. You could also have a trend in the input of methane, again allowing a potential water trend.*

  We are referring here to the trend in potential water, not then individual CH4 and H2O trends. A trend in the H2O or CH4 input would indeed result in a corresponding potential water trend, but then potential water would not be conserved (unless both trends balance, which is not expected for tropospheric trends). On the other hand, if potential water is conserved, there should be no trend in potential water.

  For clarification, we will reformulate this sentence:

  **If potential water is conserved, the potential water trend should be zero.**

- *Page 12: I really don't understand the point of this sentence "Considering this error, the combined trend above about 20 km is in a statistical sense not significant, meaning that the assumption that all water vapour is produced from methane via the net reaction (R2) is not disproved by the measurements." One should keep in mind that all water vapor is not produced from methane (ie, the average entry value is on the order of 3.5 ppmv, current methane is ~1.8 ppmv, so if all were oxidized you could get a contributions of 3.6 ppmv, so at most you could get half of water vapor from methane. It may be that here the authors are trying to assess contribution to the trend. Rohs et al, 2006, JGR, determined for the 78-03 trend in stratospheric water vapor, only 25% can be due to a trend in methane. A similar analysis could be done here, for the SCIAMACHY period.*

  Indeed, since we are looking at anomalies, we refer here to the changes of water vapour and methane, i.e. stratospheric production/loss. As suggested by referee #1, this sentence will be changed to:

**Given that the trends in potential water between 21 and 45 km lack statistical significance, there is no evidence that water vapour is produced in the stratosphere by any mechanism other than methane oxidation.**

The analysis of Rohs et al. requires as input in addition to stratospheric $CH_4$ and $H_2O$ trends also the tropospheric $CH_4$ trends and information about age of air. It is not possible to derive tropospheric trends and age of air from our data, and the stratospheric trends we derive are very small and often not significant (as are the tropospheric trends during this time period). Therefore we think it is not reasonable to include results from such an assessment in the manuscript.

- *Page 14, line 26-30: this description of the processes going on is in error. In the upper altitudes, water vapor changes are anti correlated with methane, and simply reflect age of air variations; the QBO signal is not "carried by methane".*

This paragraph has been reformulated for clarification (see also comments of other referees):

**The QBO signal is observed in both methane and water vapour at higher stratospheric altitudes. In contrast, the tropospheric methane entering the stratosphere via the lower branch of the Brewer-Dobson circulation is not impacted by the QBO at lower altitudes. The QBO signature in the upper altitude data can be explained by a QBO-dependent modulation of the transport to higher latitudes via the deep branch of the Brewer-Dobson circulation, similar to the variation in tropical aerosol extinction coefficients as seen by Brinkhoff et al. (2015) at 30 km.**

---

## Author Response (AR2)

**Water Vapour and Methane Coupling in the Stratosphere observed with SCIAMACHY Solar Occultation Measurements**

by S. Noël et al.

MS No.: acp-2017-893

**Authors' Response**

In the following, the original reviewer's comments are given in *italics*, our answer in normal font and the proposed updated text for the new version of the manuscript in **bold** font.

**Reply to report 1 from Referee 2**

**General overview**

*I don't feel the authors have adequately responded to the reviewer's comments. Presenting the new data set and validation would make a fine paper, the added science related to trends and variability has flaws, detailed comments are given below.*

We thank the reviewer for his thorough set of comments. We argue that our study is more than a new data set and a validation paper. We have analysed the data sets of CH4, H2O and potential water as function of altitude in the stratosphere for the range of latitudes observed by the solar occultation mode of SCIAMACHY. We report on the results of these analyses. We consider that these analyses shed light on the behavior of CH4 and H2O in the stratosphere in the period 2003 till 2011. These cover the full years of observations. Only few stratospheric data sets for these species exist. We agree that some of the formulations used especially in chapters 3.4, 3.5 and 4 are still unclear and therefore appreciate the additional comments and suggestions by the reviewer, which we have used to improve the paper.

**Detailed comments**

1. *The abstract states: "A significant positive water vapour trend for the time 2003–2011 is observed at lower stratospheric altitudes with a value of about 0.015 +/-0.008 ppmv year around 17km." As noted in my previous review, the time series under consideration is not long enough to talk about trends. If you look at Figure 7 and Figure 6 and Figure 8, it is clear that the end point in the lower stratosphere is a large positive anomaly (see 17 and 25 km), so that any so-called trend calculated is really a consequence of the endpoints used. If data collection had continued for a few more years, the trend wouldn't be there. If your endpoint was in 2010, you would calculate the opposite trend. You may also have trends induced by the variation in latitude in the time series. From Figure 6, it looks like you start with a latitude around 50N, and ends at 60N; that will induce an age change, which may reflect different entry conditions.*

   We agree with the referee that the time series is too short for long-term trend estimates. Therefore we explicitly mention (in reply to the previous review) at the begin of the trends section:

   "The time series of SCIAMACHY data covers only ten (nine complete) years. Consequently it is not possible to derive from these data long-term trends."

   As this still seems to be misleading, we use the term "linear changes" in the updated manuscript instead of "trend" to make clear that we are not referring to long-term climatological trends. The nomenclature in Fig. 11 has also been changed accordingly.

   The referee is also right, that the linear changes we derive are very specific for the time interval. This is why we explicitly state "for the time 2003–2011". This is especially true for the H2O and CH4 changes, where year-to-year variations are quite large. However, potential water changes have shown to be less sensitive to changes of the time interval.

We have used a bootstrap method (see e.g. Efron, 1979, `https://doi.org/10.1214/aos/1176344552`) to to investigate the sensitivity of our results to the chosen time interval. For this, we constructed for each altitude a set of 100 time series of same length as the original 2003–2011 time series (108 months) but consisting of randomly chosen combinations of results from individual months. For each of these artificial time series a linear change has been determined. For the resulting changes we determined for each altitude the mean linear change and the sample standard deviation. These values agree almost perfectly with the linear changes and their reported errors, respectively, for the complete time series (see Figure below; could be included as supplementary information, if required).

[Figure]

This shows, that our results are robust within the given errors. We therefore conclude that although the actual values of the derived linear changes vary depending on the time interval in the fit our main conclusions remain valid. We have mentioned this in the updated manuscript.

The linear changes of course also depend on latitude. In the case of SCIAMACHY solar occultation this is especially important because of the direct coupling between latitude and time within one year. However, due to the sun-fixed orbit of ENVISAT, this relation is the same every year (see top panel of Fig. 6). As the variation in the latitude is strictly coupled with the season and is the same for all years of SCIAMACHY operation its influence is removed to a large extend when calculating anomalies. However, some minor effects resulting from different trends at different latitudes might remain. Furthermore, we use only complete years in the analysis, which should also reduce the impact of this coupling. Nevertheless, there is a dependence on the specific spatial and temporal sampling of the measurements which cannot be avoided. We clarify this in the updated text.

2. *Abstract also says "Above about 20km most of the water vapour is attributed to the oxidation of methane." What you should say is that, above 20 km, most of the addition of water vapour is due to methane oxidation. Parcels enter the stratosphere with a water vapour mixing ratio between 3 and 6 ppmv. Methane enters with something around 2 ppmv. If all methane were oxidized, that would give 4 ppmv additional, but it isn't all oxidzed. Your figure 3 shows a peak stratospheric value of about 7 ppmv. Therefore, you may be able to get away with saying that above 20 km, up to have of the water vapour present can be attributed to methane oxidation.*

The referee is right, we change the sentence in the abstract to:

**Above about 20 km most of the additional water vapour is attributed to the oxidation of methane.**

3. *I also don't think this statement is correct "Further, the increasing methane input into the stratosphere due to the rise of tropospheric methane after 2007 may have contributed to the increased water vapour in the extratropical lower stratosphere as observed by SCIAMACHY." What is the "lower stratosphere"? Age of air in the extratropical stratosphere (if we're thinking below 70 mb) is on the order of months. Very little methane oxidation has occurred there (you have a small contribution*

*due to air that has descended from high up in the upper branch of the BDC), so there can't be much measurable increase in water due to that. You need to look to variations in tropical temperature to understand lower stratospheric water vapour variability.*

Also here, the referee is absolutely right. This sentence is simply wrong, we refer here to the change of potential water, not water vapour (see also Discussion). However, as also written in the discussion, the rise of tropospheric methane can only partly explain the increase in potential water. We therefore decided to remove this sentence from the abstract.

4. *Introduction: you should just delete the first 3 sentences, they add nothing to this paper. Start with "Water vapour and methane play an important role in a the chemistry of the stratosphere."*

   Agreed.

5. *Page 2, paragraph starting with "The vast majority of...." Delete the first 2 sentences of this paragraph, they are not relevant to the paper. And change "Water vapour enters..." to "The stratospheric entry value of water vapour is set by processes in the tropical tropopause layer (TTL). Also, the statement about the hygropause is not entirely correct. In the tropics, the level of minimum water varies with season (that is the whole point of the tropical tape recorder papers by Mote et al.) Sometimes it is 2 km above the tropopause in the tropics. In the mid latitudes, the level of the hygropause is a function of horizontal transport from the tropical cold point, so it will be elevated relative to the extratroipcal tropopause.*

   The text has been changed as proposed and the additional information about the hygropause has been added:

   **A minimum in water vapour, which is around 2 km above the tropopause in the tropics, is called the hygropause. The level of minimum water varies with season and latitude. In the mid-latitudes, the level of the hygropause is a function of horizontal transport from the tropical cold point, so it will be elevated relative to the extratropical tropopause.**

6. *You don't need figure 1 in this paper.*

   We agree that this figure is not absolutely necessary, but as mentioned in our previous reply we think that this figure might be useful for the less experienced reader as it illustrates the basic transport pathways and terminology. Therefore we prefer to keep this figure in the paper.

7. *Paragraph starting with line 16: The amount of methane is going to vary with how the tropospheric burden varies. Water vapour will not; it will vary due to tropical tropopause temperature changes, and possible changes in monsoon circulations, mixing in from midlatitudes, convection, microphysics, etc. This sentence "The Brewer-Dobson circulation controls the tropical upwelling." is also not quite valid. The BDC may reflect tropical upwelling, it doesn't control it. Note, the description of the driving of the BDC is valid for the upper branch, it's a much more complicated situation for the lower branch. Also note, the TIL is really irrelevant here, just delete that sentence.*

   We reformulate this section:

   **The amount of methane entering the stratosphere in the tropics depends on the changing strength of the sources (e.g. possible tropospheric trends). Variations in water vapour are caused by tropical tropopause temperature changes and dynamical effects, like changes in monsoon circulations, mixing in from mid-latitudes, convection, and microphysical processes. Furthermore, variations in the Brewer-Dobson circulation (on seasonal and inter-annual time scales) and the Quasi-Biennial-Oscillation (QBO, see e.g. Baldwin et al. (2001); Butchart (2014), and references therein) play a role.**
   **The Brewer-Dobson circulation in the upper branch is driven by middle latitude planetary waves entering the stratosphere and as a consequence leads to adiabatic cooling in the tropical UTLS (upper troposphere / lower stratosphere region) related to the increased upwelling which strongly determines the stratospheric entry of water vapour in the tropics (Randel et al., 2006; Dhomse et al., 2008).**
   **As a component of the Brewer-Dobson circulation, the tropical upwelling is resposible for the**

**transport of air masses from the troposphere into the stratosphere (both water vapor and methane) and influences the freeze-drying, i.e. the process through which the tropopause acts as a cold trap such that water vapour partly freezes out before reaching the stratosphere (e.g. Fueglistaler and Haynes, 2005).**

The sentence on TIL has been deleted.

8. *Line 30, I would rewrite to say "Water vapour production in the stratosphere is largely a consequence of methane oxidation."*

   Has been changed.

9. *Page 3: discussion of potential water: the authors should note that 2CH4+H2O is effectively conserved following a stratospheric parcel, not in the stratosphere. For example, if you look at distributions from an Eulerian rather than a Lagrangian standpoint, you'll see variations related to variations in water vapour entry...ie, the seasonal cycle tape recorder, variations related to the QBO impacting lower stratospheric temperatures and mixing.*

   Agreed.

   The section has been changed to:

   **For this overall reaction one methane molecule finally produces two water vapour molecules, which means that the sum of volume mixing ratios $[H_2O] + 2[CH_4]$, referred to as potential water (PW), see e.g. Rinsland et al. (1996); Nassar et al. (2005) and references therein, is expected to be roughly conserved following a stratospheric parcel. Since the actually conserved quantity is total hydrogen, this assumes that variations in $H_2$ can be neglected. The latter is in fact not always the case, as investigations by e.g. Juckes (2007) and Wrotny et al. (2010) indicate. Furthermore, potential water at a certain altitude may be affected by variations in the water vapour entry or QBO impacting lower stratospheric temperatures and mixing.**

10. *Page 3, line 6...delete "probably"*

    Done.

11. *Page 3, line 9...Age of air is a function of altitude and latitude and season. 2 years at 15 km, even at mid latitudes is too long. Note recent work by Ray et al. (http://onlinelibrary.wiley.com/doi/ 10.1002/2016JD026198/abstract) detailing mesospheric sinks of SF6 that can contaminate polar age of air estimates.*

    We agree that there is some uncertainty in the age of air values especially at polar latitudes and added the following sentence and a reference to Ray et al.:

    **Age of air depends on season and latitude, and recent work by Ray et al. (2017) indicates, that these age estimates might be too high especially inside the polar vortex.**

12. *Page 3, line 12. This statement is wrong "However, the mixing of air masses during transport does not affect the total hydrogen balance such that potential water should still be conserved." That depends on whether you are mixing air masses with the same value of 2CH4+H2O. Consider at the edge of the subtropical, near 400K in January. The tropical 2CH4+H2O will be at an annual minimum. On the poleward side of the jet, you'll have air that entered a few months earlier with warmer tropical tropopause temperatures, and that value of 2CH4+H2O will be larger. Mixing of those will not "conserve" potential water, but it will conserve the sum of the potential water.*

    Agreed. We have removed this sentence.

13. *Page 3 line 16. The paper states "Ideally, both water vapour and methane should be retrieved from measurements by the same instrument." I see no reason why this is true. You want both water vapour and methane to be accurate, and co-located, but they don't have to be from the same instrument.*

    Agreed, the main point is indeed that the data are accurate and collocated. However, with the same instrument, especially collocation is usually much easier achieved than with different sensors on

different platforms, and possible cancellation of systematic H2O and CH4 errors might improve the final PW.

To clarify this we modify the text as follows:

**For this, both water vapour and methane data should be collocated and accurate. If the underlying measurements are from the same instrument, the collocation of the two data sets is usually very close. Furthermore, possible systematic errors in methane or water vapour caused e.g. by instrument calibration or by the retrieval method may to some extent cancel for potential water.**

14. *Figure 7 caption...make it clear that you are looking at varying latitudes in this plot (perhaps refer back to latitude plot in figure 6)*

We have added to the caption:

**Note that for these data the same latitudinal dependence as shown in Fig. 6 applies.**

15. *Figure 7: Does the satellite coverage give the same latitude for the same day in the year for each year? If it doesn't, I'm not sure what the anomalies actually mean.*

Yes, due to the sun-fixed orbit there is a direct relation between latitude and time of the year (same for each year), as explained in the text and shown in the top panel of Fig. 6.

16. *Page 11, line 9, the age of air at 30 km is only calculated to be 8 years if you use the MIPAS SF6 derived ages. If you look at the Haenel paper plots of balloon derived ages, they are significantly less. Also, the larger than a 2-year shift doesn't make sense. Any variations you're seeing that are QBO related in the high latitudes are largely going to be dynamical (and conservations of mass related). There could possibly be some entry value related signal due to QBO induced tropical tropopause temperature changes, but those should be fairly small, and mixed out by the time air has reached an age in the range of 4-8 years. I recommend deleting this whole discussion.*

Agreed. We have reformulated the sentence to:

**The phase shift between stratospheric wind and SCIAMACHY data is caused by various dynamical processes during the transport of air from the tropics (where Singapore winds are measured) and the mid/high latitudes of the SCIAMACHY data and cannot be determined well from our 9-year time series.**

and have removed the following sentences:

"Note that the age of air at these altitudes may be up to about 8 years according to e.g. Haenel et al. (2015). Consequently, the actual phase shift is expected to be larger than one 2-year period of the QBO. It therefore cannot be determined well from our 9-year time series."

17. *Page 12: potential water discussion: 2CH4+H2O will be conserved in a Lagrangian sense, not a Eulerian sense. Your measurements are effectively Eulerian, so I'm not convince this discussion is meaningful.*

In principle the referee is right, the conservation of PW is only guaranteed in a Lagrangian sense, i.e. for an air parcel. However, this would also be the case in the Eulerian sense if transport (i.e. BDC) and sources are constant. So, if we measure a conservation of PW this is an indication for a constant transport and changes, and if we measure changes in PW this is an indication for changes in transport or sources - this is just what we write in the first paragraph of this section. As we indicate in the manuscript, in the case of changing PW it is difficult to distinguish between both reasons (without additional information from e.g. tropical measurements). Nevertheless we think it is useful to look at changes of PW.

18. *Page 13, trends section. The time series isn't long enough to do trend analysis, and the variations of sampling with latitudes make this really complicated, and that isn't properly acknowledged. I recommend deleting this entire section. In particular, potential water is conserved in a parcel, thereby*

*giving a potential water trend of zero. But, this data set isn't looking at parcels, so the whole discussion doesn't make sense. I agree that water vapor is only produced in the stratosphere by methane oxidation, but I don't think this analysis proves that.*

As mentioned in our answer to the first point above, we updated the text to clarify that we determine linear changes for a specific time interval and spatial/temporal sampling. As mentioned above, the latitudinal sampling effect should be largely reduced by using anomalies. More critical is the length of the time series, but as described above we think that the derived changes for PW are robust and support our main conclusions. W.r.t. conservation of PW in a parcel, see our answer to the previous point. We agree that we cannot prove with our measurements that water vapour is only produced in the stratosphere by methane oxidation, but we show that this assumption is not in contrast to our data.

19. *Discussion...methane, QBO, and BDC are not the only factors impacting stratospheric water vapour. ENSO has been shown to play a role, SSWs can play a role, I'm not sure what the authors mean by "varying tropical tropospheric water vapour", but I would expect that would only play a role if you've changed the fraction of air that bypasses the cold point (via overshoots), or that has actually changed the tropical tropopause temperature radiatively.*

The formulation "varying tropical tropospheric water vapour" is indeed misleading. We do not mean changes in tropospheric water vapour, but variations in the entry of tropospheric water vapour into the tropical stratosphere due to various effects, including the ones mentioned by the referee.

We have reformulated this as follows:

**It is therefore likely, that potential water at lower stratospheric altitudes is influenced by variations in the entry of tropospheric water vapour into the tropical stratosphere.**

20. *Last paragraph on page 15: I think this needs to be stated more clearly. What do you mean by "water vapour changes"? Is this in a Lagrangian sense (ie, the only way you change water vapour in a parcel is via production due to methane oxidation...and effectively that is correct if you're ignoring molecular hydrogen), or are you considering variation at a given location (in an Eulerian sense), in that if water increases, you're looking at air that is effectively older, so it has a larger amount of water (and lower amount of methane.) I also don't think the statement "Above 20 km , in the region of the deep branch of the Brewer-Dobson circulation, air is older. This enables oxidation of methane to water vapour to be completed rapidly." is correct. It is not the fact that the air is older that allows the methane oxidation, it's a function of the lifetime relative to altitude (see LeTexier et al, QJRMS fig 1, and it should probably be 30 km instead of 20 km).*

With "changes" we mean the differences to the amount of water vapour injected in the tropics. As our measurements are effectively Eulerian, we are referring to changes at a certain altitude (as we write it: "at these altitudes water vapour changes ..."). Depending on the age of air the ratio between methane and water vapour is different, but if potential water is conserved, this is an indication that (additional) water vapour is mainly produced by the oxidation of methane. We have clarified this in the updated text. We have also reformulated the sentence "Above 20 km..." according to the suggestions of the referee.

Updated section:

**In the upper dynamical regime water vapour is produced from methane oxidation and potential water anomalies are to a good approximation homogeneous with altitude and change on longer time scales. In the region of the deep branch of the Brewer-Dobson circulation, photochemical lifetimes decrease with altitude (LeTexier et al., 1988). This enables oxidation of methane to water vapour to be completed rapidly. As a result variations of both gases are in phase and potential water is essentially conserved (Fig. 8). Consequently at these altitudes water vapour changes (relative to the injected amounts in the tropics) can be concluded to be mainly determined by the oxidation of methane.**

21. *First paragraph page 16, delete this paragraph, you don't have the length of time series to be considering 5–6 year oscillations.*

In response to previous comments, we do not talk about oscillations here – this indeed can not be shown with our short time series. What we observe (and write) is "a change of potential water at higher altitudes on a timescale of 5–6 years". This could be a specific feature for the time period of the SCIAMACHY data. We have updated the text to clarify this by specifying the time interval 2003–2011. We have also adapted the wording in the abstract accordingly.

**Reply to report 2 from Referee 3**

Note: It seems that these comments refer to a previous version of the manuscript (i.e. the ACPD version, not the revised version).

**General comment**

*The changes made to the manuscript have improved the presentation, but there remain a few points which should be changed.*

**Specific comments**

1. *The abstract remains unchanged and confusing. First, in line 10, it would be helpful to replace "correlated" with "anticorrelated". But most confusing is the sentence starting in line 11: "Above about 20km most of the water vapour seems to be produced by methane, but short-term fluctuations and a temporal variation on a scale of 5–6 years are observed." The first half of this sentence is okay, but the second half of this sentence should certainly not follow a "but". Perhaps all that is required is to make this second phrase a new sentence.*

   In the revised version the requested changes are already included.

2. *Page 15 line 18 – "Altitudes above about 20km are fed via the deep branch of the Brewer-Dobson circulation, and water vapour is essentially produced from methane oxidation." This sentence was in the last version as well, but I failed to notice it. The word "essentially" is far too strong here, since it implies that methane oxidation produces the vast majority of H2O in the region. This should be rephrased as "most water vapour is produced from methane oxidation".*

   Actually, also this section was already updated in the revised version.

3. *Page 15 line 21 – "possibly in combination with changes of water vapour" should be "possibly in combination with changes in water vapour entering the stratosphere". Or, it would be even better if the authors would simply rewrite this last sentence as "The rise of tropospheric methane after 2007 reaches these lower stratospheric altitudes with a delay of about 2 years, contributing to the observed increase of potential water after 2009."*

   A similar change was already included in the revised version:

   **The increase of tropospheric methane after 2007 reaches these lower stratospheric altitudes with a delay of about 2 years. This contributes in part to the observed increase of potential water after 2009, but additional processes such as changes of tropospheric water vapour input are required for a quantitative explanation.**

**List of changes**

Changes according to the comments mentioned above have been made in the revised manuscript. Furthermore, the wording of the title has been slightly changed and some typos have been corrected. The changes are marked in the attached version of the revised manuscript.

[revised manuscript text omitted]